# Neuropathological and transcriptomic characteristics of the aged brain

Jeremy A Miller[1], Angela Guillozet-Bongaarts[1], Laura E Gibbons[2], Nadia Postupna[3], Anne Renz[4], Allison E Beller[3], Susan M Sunkin[1], Lydia Ng[1], Shannon E Rose[3], Kimberly A Smith[1], Aaron Szafer[1], Chris Barber[1], Darren Bertagnolli[1], Kristopher Bickley[1], Krissy Brouner[1], Shiella Caldejon[1], Mike Chapin[1], Mindy L Chua[3], Natalie M Coleman[3], Eiron Cudaback[3], Christine Cuhaciyan[1], Rachel A Dalley[1], Nick Dee[1], Tsega Desta[1], Tim A Dolbeare[1], Nadezhda I Dotson[1], Michael Fisher[1], Nathalie Gaudreault[1], Garrett Gee[1], Terri L Gilbert[1], Jeff Goldy[1], Fiona Griffin[1], Caroline Habel[1], Zeb Haradon[1], Nika Hejazinia[1], Leanne L Hellstern[3], Steve Horvath[5], Kim Howard[3], Robert Howard[1], Justin Johal[1], Nikolas L Jorstad[3], Samuel R Josephsen[3], Chihchau L Kuan[1], Florence Lai[1], Eric Lee[1], Felix Lee[1], Tracy Lemon[1], Xianwu Li[3], Desiree A Marshall[3], Jose Melchor[1], Shubhabrata Mukherjee[2], Julie Nyhus[1], Julie Pendergraft[1], Lydia Potekhina[1], Elizabeth Y Rha[3], Samantha Rice[3], David Rosen[1], Abharika Sapru[3], Aimee Schantz[3], Elaine Shen[1], Emily Sherfield[3], Shu Shi[1], Andy J Sodt[1], Nivretta Thatra[1], Michael Tieu[1], Angela M Wilson[3], Thomas J Montine[3], Eric B Larson[4], Amy Bernard[1], Paul K Crane[2], Richard G Ellenbogen[6], C Dirk Keene[3†], Ed Lein[1†*]

[1]Allen Institute for Brain Science, Seattle, United States; [2]Department of Medicine, University of Washington, Seattle, United States; [3]Department of Pathology, University of Washington, Seattle, United States; [4]Kaiser Permanente Washington Health Research Institute, Seattle, United States; [5]Department of Human Genetics, University of California, Los Angeles, Los Angeles, United States; [6]Department of Neurological Surgery, University of Washington, Seattle, United States

*For correspondence: EdL@alleninstitute.org

†These authors contributed equally to this work

**Abstract** As more people live longer, age-related neurodegenerative diseases are an increasingly important societal health issue. Treatments targeting specific pathologies such as amyloid beta in Alzheimer's disease (AD) have not led to effective treatments, and there is increasing evidence of a disconnect between traditional pathology and cognitive abilities with advancing age, indicative of individual variation in resilience to pathology. Here, we generated a comprehensive neuropathological, molecular, and transcriptomic characterization of hippocampus and two regions cortex in 107 aged donors (median = 90) from the Adult Changes in Thought (ACT) study as a freely-available resource (http://aging.brain-map.org/). We confirm established associations between AD pathology and dementia, albeit with increased, presumably aging-related variability, and identify sets of co-expressed genes correlated with pathological tau and inflammation markers. Finally, we demonstrate a relationship between dementia and RNA quality, and find common gene signatures, highlighting the importance of properly controlling for RNA quality when studying dementia.
DOI: https://doi.org/10.7554/eLife.31126.001

## Introduction

The population of the United States is aging, with the fastest growth in the very oldest part of the population where the number of nonagenarians and centenarians are expected to increase from 2 million to 10 million by 2050 (*Corrada et al., 2012*). This creates a significant public health challenge due to the increased health issues related to age, most notably the debilitating effects of neurodegenerative diseases. Dementia is thought to affect 11% of the US population over the age of 65 overall, and the incidence of dementia onset roughly doubles every five years to ~40–60% after age 90 (*Corrada et al., 2012*; *Gardner et al., 2013*; *Alzheimer's Association, 2016*). Approximately 2/3 of dementia cases are diagnosed as Alzheimer's disease (AD), representing an estimated 13.8 million cases by 2050 (*Alzheimer's Association, 2016*), with most of the remaining cases diagnosed as vascular dementia or mixed dementia from multiple etiologies. AD is characterized by stereotyped progressive neurodegeneration and accumulation of two misfolded proteins in brain regions important for cognition and memory. Hyperphosphorylated tau is thought to form intracellular neurofibrillary tangles (NFTs) initially in projection neurons of the entorhinal cortex, which then spread to the hippocampus and neocortex (*Braak and Braak, 1991*). Similarly, Aβ forms into extracellular plaques in cortical and deep brain structures (*Mirra et al., 1991*). In addition, Lewy bodies can be identified in cerebral cortex and deep nuclei as well as brainstem, and microvascular lesions can occur throughout the brain. While diagnostic grading systems for these pathologies have been developed that variably correlate with cognitive and behavioral function, there is no consensus on whether these microscopically observed disease pathologies are causal or effects of other underlying processes. Despite enormous efforts, no anti-tau or anti-amyloid therapies have been successful, and those limited treatment options targeting symptoms are based on acetylcholine or NMDA metabolism (*Allgaier and Allgaier, 2014*).

To further complicated diagnosis and treatment, pathologies associated with dementia are widespread in the aged brain even in the absence of dementia; for example, nearly half of non-demented participants in the 90+ Autopsy Study met pathological criteria for AD (*Corrada et al., 2012*). Among individuals with dementia, pathological phosphorylated tau (pTau) and amyloid beta (Aβ) pathology findings are actually lower in the 90+ year old age group than in the 60–80 year old group (*Haroutunian et al., 2008*), whereas other neuropathological conditions such as Lewy bodies and hippocampal sclerosis were only identified in individuals with dementia (*Corrada et al., 2012*). Indeed, while the overall incidence of dementia increases with age, pathology becomes much more variable, the relationship between disease pathologies and cognition weakens (*Haroutunian et al., 2008*; *Corrada et al., 2012*), and the relevance of canonical risk factors for AD, including *APOE* genotype, decreases with advancing age (*Gardner et al., 2013*). Identification of biological and environmental factors critical to the etiology and progression of neurodegenerative processes will be critical to developing preventive and therapeutic strategies in the aged brain.

Genome-wide gene expression analyses have been applied to identify molecular pathways affected by aging and dementia. Transcriptomics shows robust and stereotyped gene expression patterning in the brain, including spatial (brain region) (*Hawrylycz et al., 2012*) and temporal variation over the lifespan from development through adulthood into aging (*Colantuoni et al., 2011*) (http://brainspan.org). The aged brain shows increased variability in this transcriptional patterning compared to younger brains (*Colantuoni et al., 2011*). Comparing brains of people who died with a clinical diagnosis of AD to brains of people who died with no dementia, a number of studies have identified dysfunction of pathways and biological processes including synaptic transmission, energy metabolism, inflammation, cytoskeletal dynamics, signal transduction, transcription factors, and cell proliferation (*Colangelo et al., 2002*; *Blalock et al., 2004*; *Webster et al., 2009*; *Miller et al., 2013*). Many of these same pathways show disrupted gene expression in older compared with younger individuals not diagnosed with dementia (*Miller et al., 2008*), although in many cases to a lesser extent and in different brain regions (*Avramopoulos et al., 2011*). Since many of the gene expression studies to date focus on somewhat younger cohorts and have only limited information on disease pathologies, it is unclear whether robust relationships between gene expression and disease pathology or cognition extend to older individuals.

To better understand the relationship between cognition, brain pathology and injury, and gene expression in the aged brain, we created the Aging, Dementia, and Traumatic Brain Injury (TBI) Study, which is a detailed neuropathological, molecular, and transcriptomic characterization of brains

of 107 people from the Adult Changes in Thought (ACT) cohort. The ACT study was designed as a population-based, prospective study of normal brain aging and dementia, and incorporates extensive medical history and postmortem characterization (*Kukull et al., 2002*; *Larson et al., 2006*; *Crane et al., 2013*). ACT participants entering the study are at least 65 years old and free of dementia, and the median age at death of the cohort used for this study is 90. This freely available resource (http://aging.brain-map.org/) presents a systematic and extensive dataset of study participant metadata, quantitative histology and protein measurements of neuropathology, and RNA sequencing (RNA-seq) analysis of hippocampus and neocortex. Here we describe this resource and initial analyses to understand features of the aged brain, the relationship between dementia and pathology, and transcriptional signatures of dementia, neuropathology and aging. Code and additional files required to reproduce all analyses are available in Github (https://github.com/AllenInstitute/agedbrain; *Miller, 2017*; copy archived at https://github.com/elifesciences-publications/agedbrain).

## Results

### A multimodal atlas of aging and dementia

The Aging, Dementia, and TBI Study was initially designed to study the long term effects of mild-to-moderate TBI, but we focus the current analysis on aging and dementia and present our results with respect to TBI elsewhere. This study includes 55 participants of the ACT study self-reporting TBI with loss of consciousness, along with 55 individuals matched for age, sex, and year of death who did not report a TBI with loss of consciousness. Donors in the exposure cohort reported between 1–3 lifetime TBIs with loss of consciousness ranging from <10 s to >1 hr (*Figure 1A*). Most participants were male (63 males, 44 females), with a wide range of educational backgrounds, and quite old (77–102 years old at time of death, median = 90), representing one of the oldest cohorts of its kind to date. Around half of the donors were diagnosed with dementia, including 30 with AD, 12 with dementia of multiple etiologies, and four with vascular dementia. More *APOE* ε4-positive participants had dementia (65%) than *APOE*ε4 negative participants (40%), consistent with the role of this

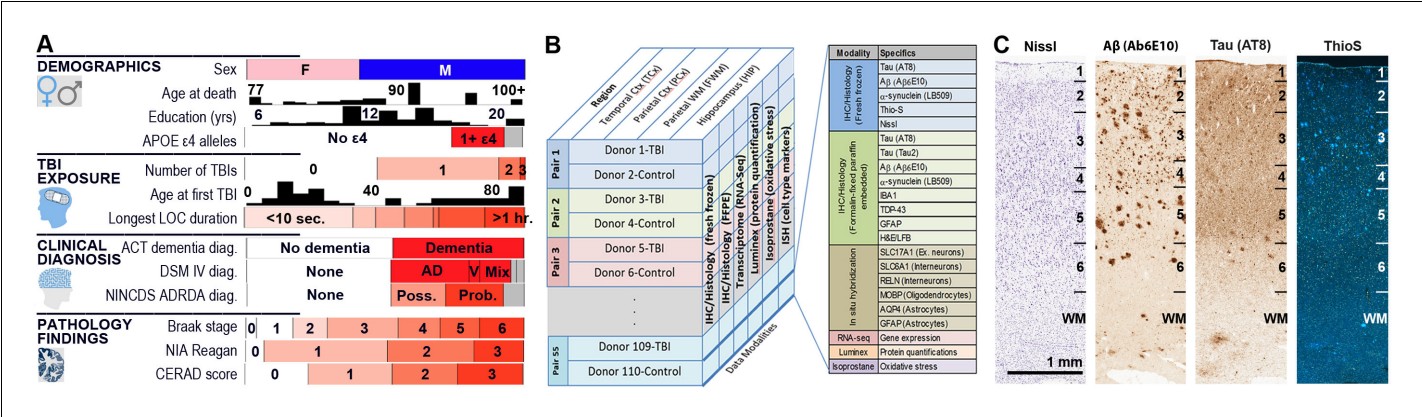

**Figure 1.** Experimental design and cohort characteristics. (**A**) Demographics for all 106 donors (after excluding one outlier; Materials and methods). Histograms are shown for Age at death, Education (yrs), and Age at first TBI. All other metrics (except sex) are sorted from lowest to highest, with white corresponding to none (or 0 or control), and red corresponding to the highest severity of the condition or pathology. (**B**) Summary of all data available for each donor included in this study, including IHC on fresh and frozen tissue, RNA-seq analysis, and Luminex protein and isoprostane quantification. (**C**) Examples of histology from fresh frozen temporal cortex of donor H14.09.075 using IHC for Ab6E10 and AT8 and ThioS labeling, showing severe Ab and pTau pathology. Numbers indicate cortical layers. Descriptions of each metric (including abbreviations used) are included as a downloadable file on http://aging.brain-map.org/download/index. See also *Figure 1—figure supplements 1–2*.
DOI: https://doi.org/10.7554/eLife.31126.002

The following figure supplements are available for figure 1:

**Figure supplement 1.** Example of FFPE pathology in donor H14.09.075.
DOI: https://doi.org/10.7554/eLife.31126.003

**Figure supplement 2.** Dominant gene expression signatures in RNA-Seq data are brain region and RIN.
DOI: https://doi.org/10.7554/eLife.31126.004

gene as a primary genetic risk factor for AD. Deposition of the disease pathologies pTau, in the form of NFTs (Braak stage), and Aβ, in the form of neuritic plaque density (CERAD score), ranged from absent through severe with relatively equal frequency (*Figure 1A*), and was generally higher in donors with dementia compared to donors without dementia, as expected (*Table 1*). It should be noted that this cohort is not representative of the ACT cohort as a whole (i.e., it is older, more heavily male, and all deceased). Analyses can be extrapolated back to the entire ACT cohort using weights (*Haneuse et al., 2009*) available on the online resource (http://aging.brain-map.org/download/index); however, since we did not observe substantially different results using these weights (data not shown) we choose not to use them in the analyses presented here.

For each donor, we collected tissue from four brain regions known to show neurodegeneration and pathology as a result of AD and Lewy body disease (LBD; hippocampus and temporal and parietal cortex) (*Braak and Braak, 1991*; *McKeith et al., 1996*; *Hyman et al., 2012*; *Montine et al., 2012*), hippocampal sclerosis and phospho (p) TDP-43 pathology (hippocampus and temporal cortex) (*Nelson et al., 2016*), chronic traumatic encephalopathy (CTE; temporal cortex and parietal cortex and white matter) (*McKee et al., 2016*), and microvascular brain injury (multiple regions) (*Flanagan et al., 2016*), and characterized each tissue in a highly standardized manner using a broad set of informative data modalities (*Figure 1B*). We used immunohistochemistry (IHC) on both fresh frozen and formalin fixed paraffin embedded (FFPE) tissue to stain and quantify proteins marking dementia-related pathologic findings, including pTau, Aβ, α-synuclein (Lewy bodies), and pTDP-43, as well as microglia (IBA1) and astrocytes (GFAP). For example, donor H14.09.075 (78 year old female with dementia) shows significant pathology of intracellular pTau (AT8) and extracellular Aβ plaques (Ab6E10 and ThioS) based on IHC of fresh frozen (*Figure 1C*) and fixed (*Figure 1—figure supplement 1*) tissue. Negligible amounts of α-synuclein were observed in these brain regions. In addition, we used multiplexed Luminex assays for protein molecular quantification of tau and pTau variants and Aβ species, as well as for α-synuclein, inflammatory mediators (cytokines and chemokines), neurotrophic factors, and other targets. We determined free radical injury in parietal and temporal cortex using GC/MS quantitation of isoprostanoids. We used in situ hybridization (ISH) to detect expression for canonical marker genes for astrocytes (*AQP4*, *GFAP*), oligodendrocytes (*MOBP*), and neuronal subtypes (*RELN*, *SLC6A1*, *SLC17A7*) to provide insight into the cellular makeup of tissues used for neuropathological and transcriptomic analysis. Finally, we used RNA-Seq to assess genome-wide expression levels of >50,000 coding and non-coding transcripts on macro-dissected tissue sections from the same blocks used for fresh-frozen IHC and ISH. Brain region and sample RIN represent the largest source of transcriptional variation (*Figure 1—figure supplement 2*), as shown previously (*Preece and Cairns, 2003*; *Li et al., 2004*; *Tomita et al., 2004*; *Mexal et al.,*

**Table 1.** Summary of demographics for donors with and without dementia.

P-values for upper seven and lower two metrics are uncorrected significance values from T-tests and hypergeometric tests, respectively (*p<0.05 after Bonferroni correction for multiple comparisons). Demographic summary includes 106 donors used in analysis (Materials and methods).

| Category | Non-demented Mean | SD | Demented Mean | SD | P-value |
|---|---|---|---|---|---|
| Age at death (yrs) | 89 | 7 | 90 | 6 | 2.8E-01 |
| Education (yrs) | 15 | 3 | 14 | 3 | 4.9E-02 |
| Number of TBIs | 0.6 | 0.7 | 0.7 | 0.7 | 8.0E-01 |
| Age at first TBI | 23 | 31 | 24 | 32 | 8.2E-01 |
| Braak stage | 2.8 | 1.5 | 4.1 | 1.7 | 8.9E-05* |
| NIA Reagan | 1.4 | 0.7 | 1.9 | 0.9 | 1.8E-03* |
| CERAD score | 1.2 | 0.9 | 1.8 | 1.2 | 1.4E-02 |
| | Count | | Count | | P-value |
| Sex | 20 F / 36 M | | 23 F / 27 M | | 1.0E-01 |
| >0 APOE ε4 alleles | 47 No/7 Yes | | 32 No/13 Yes | | 2.1E-02 |

DOI: https://doi.org/10.7554/eLife.31126.005

*2006*; *Vawter et al., 2006*; *Hawrylycz et al., 2012*); therefore, we treat RNA-seq data from each brain region independently and correct for RIN. After excluding poor quality or otherwise unusable tissue, data from 377 tissue blocks across four brain regions in 107 donors are available as part of the resource.

## Widespread tau and amyloid beta pathology in the aged brain

NFTs and amyloid plaques are thought to progress in a stereotyped anatomical pattern with increased AD severity, but also appear to show a more general increased load in advanced aging (*Mungas et al., 2014*). We assessed disease pathology in this resource using two approaches. First, we used standard global metrics of disease severity by including NFT distribution (extent; i.e., Braak stage) (*Braak and Braak, 1991*) and neuritic amyloid plaque cortical density (i.e., CERAD score) (*Mirra et al., 1991*). Amyloid plaque distribution (Thal phase) (*Thal et al., 2002*) was not routinely available in the ACT study until 2012 and therefore was not included in the analysis. In addition to standard diagnostic neuropathological endpoints routinely assessed for each case, we used local measurements in the blocks used for RNA-seq. These included soluble protein (Luminex) and histological (IHC) analysis on adjacent frozen sections, as well as more standard histology on FFPE tissue sections from the same brain regions from the same or opposite hemisphere. To quantify pathology load in IHC we calculated the fraction of labeled pixels in representative regions of interest for each stain from each case, using a modification of a technique we previously developed for quantification of ISH signal (*Dang et al., 2007*; *Lein et al., 2007*). Quantitative scores are consistent with qualitative observations using this technique (*Figure 2—figure supplement 1*). Pathology quantifications based on frozen and fixed tissue are highly correlated (r = 0.78 for pTau and r = 0.67 for Aβ), although in some cases the threshold and dynamic ranges showed some variation between the two tissue preparations (*Figure 2A* and *Figure 2—figure supplement 2*). These quantified values also show regional patterns consistent with Braak stage and CERAD scores (*Figure 2—figure supplement 3*).

We observe a wide distribution of pTau and Aβ pathology loads in this aged cohort ranging from no pathology to extremely high (*Figure 2A*). As expected, pathological tau (AT8 IHC and pTau Luminex) tended to be higher in hippocampus while Aβ (Ab6E10 IHC and Aβ42 Luminex) is higher in cortex, consistent with known AD pathological distributions and progression. Both pTau and Aβ pathologies (as measured by Braak stage and CERAD score) are found more widely distributed in the brain among people who died at older ages compared to those who died at younger ages, despite the relatively compressed age range of this cohort (*Figure 2B*). We find a statistically significant relationship between dementia and pTau pathology, shown in *Figure 2C* as both an increased anatomical distribution (Braak stage) and hippocampal pathology load based on multiple protein quantification metrics (e.g., AT8 IHC). Despite the known differences in pathological signatures in different disorders of dementia, we find the same results for all dementia cases and the subset of cases diagnosed with probable or possible AD (*Figure 2—figure supplement 4*). This is likely due to the predominance of AD cases in our cohort, particularly when including donors diagnosed with dementia of multiple etiologies (only eight donors were diagnosed with vascular or other dementia with no clinical diagnosis of AD). We also observed a correlation between pTau levels and age (but not Aβ) in donors without dementia (*Figure 2—figure supplement 5*). Phrased differently, there is a significant difference in pTau pathology between donors with and without dementia in the younger (<90 years) but not the older (90 + years) donors (*Figure 2—figure supplement 6*), consistent with a general increase in pTau pathology with age (*Haroutunian et al., 2008*; *Corrada et al., 2012*). These results provide support for the idea that different pathways and progressions may be involved in pathological processes in the oldest old (i.e., individuals > 90 years old).

As part of the ACT Study, participants are assessed every other year for cognitive status using the Cognitive Abilities Screening Instrument (CASI) to determine whether further assessment for dementia is necessary (*Teng et al., 1999*). Tau is significantly correlated with an Item Response Theory version of CASI score (CASI_irt; *Figure 2D*) that accounts for the uneven distribution of CASI item difficulty levels across the ability spectrum (*Crane et al., 2008*; *Ehlenbach et al., 2010*); furthermore, local metrics of pathology (AT8 IHC) associate more strongly with cognitive metrics than does Braak stage, demonstrating the value of these quantitative metrics. Measures of Aβ pathology show nominally significant, but less robust associations with disease and cognitive status (*Figure 2B–D*,

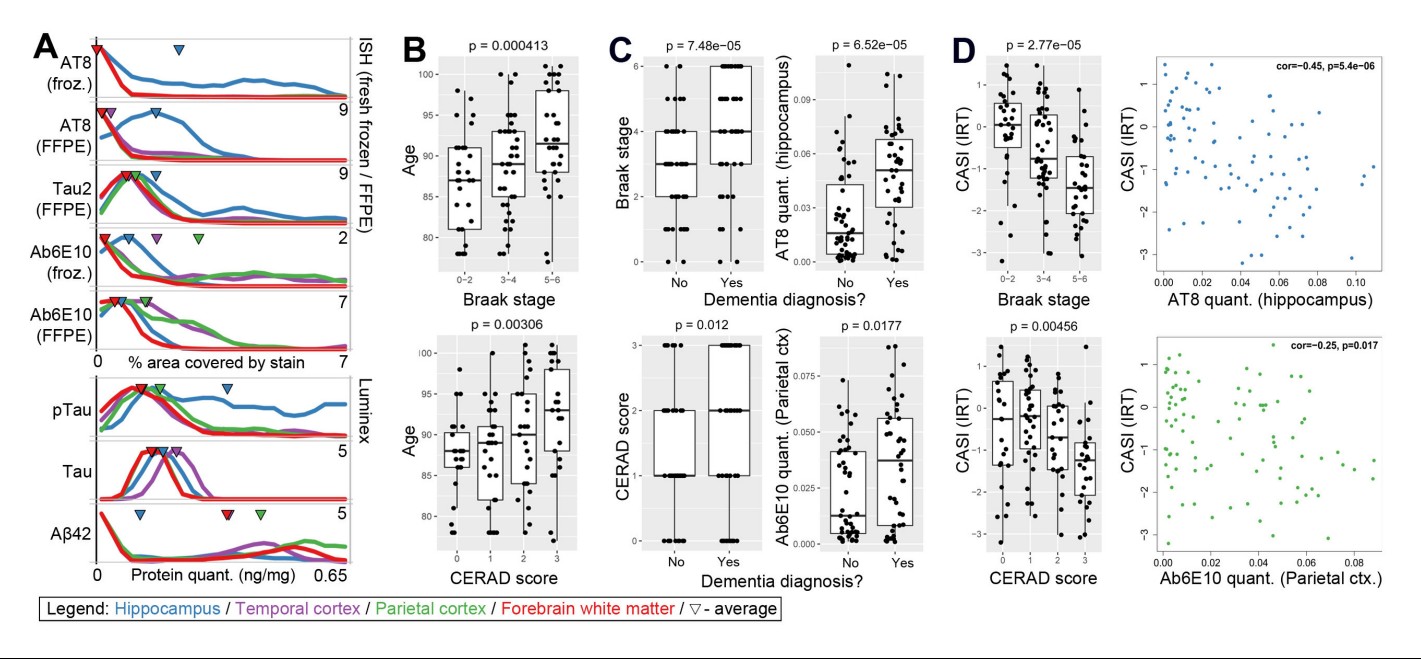

**Figure 2.** Amyloid beta and tau pathologies show a relationship with age, but with high variability. (A) Distributions of values for quantitative pathology metrics, separated by brain region (colors in legend). Lines are density plots (y-axis) of distributions of each metric (specified in x-axis label), with triangles indicating the average value. Note that several metrics have higher values in cortex than hippocampus, or vice versa. (B) Donors with higher levels of tau pathology (defined as Braak stage; top row) and of Aβ (defined as CERAD score; bottom row) were older on average (y-axis) than donors with lower measures of pathology. (C) Donors with dementia have higher levels of tau and Aβ pathology on average than donors without dementia, as measured both by global metric (Braak stage, CERAD score; left column), and local IHC quantifications in hippocampus (AT8, Ab6E10; right column). (D) Donors with higher levels of dementia pathology (x-axes; same metrics as in C) also tend to have lower cognitive scores (y-axes). For bar plots in B-D, dots indicate specific donors, and boxes and whiskers represent 25%/75% and 5%/95%, respectively. For scatterplots, dots indicate donors, with specific metrics shown on axes. See also *Figure 2—figure supplements 1–6*.

DOI: https://doi.org/10.7554/eLife.31126.006

The following figure supplements are available for figure 2:

**Figure supplement 1.** Quantitative measures of pTau and Aβ generally match qualitative observations.
DOI: https://doi.org/10.7554/eLife.31126.007
**Figure supplement 2.** Quantitative measures of pTau and Aβ are consistent between frozen and FFPE tissue.
DOI: https://doi.org/10.7554/eLife.31126.008
**Figure supplement 3.** Consistent measures of pathology across metrics.
DOI: https://doi.org/10.7554/eLife.31126.009
**Figure supplement 4.** Consistent measures of pathology for the subset of dementia donors with AD.
DOI: https://doi.org/10.7554/eLife.31126.010
**Figure supplement 5.** Tau pathology increases with age in hippocampi of non-demented donors.
DOI: https://doi.org/10.7554/eLife.31126.011
**Figure supplement 6.** Younger donors show more significant relationships between dementia status and pathology than older donors.
DOI: https://doi.org/10.7554/eLife.31126.012

bottom row), aligning with previous reports that pTau may be a better indicator of AD severity than Aβ (*Nelson et al., 2007*).

While pTau and Aβ pathologies were more common in people with dementia than cognitively normal older adults, these pathologies were highly variable across this cohort among those with and without dementia. For example, eight of the 32 donors (25%) with severe NFT pathology (Braak stage >= 5) and six of the 25 donors (24%) with severe amyloid pathology (CERAD score = 3) did not have dementia. This disconnect is dramatic in individual cases where extremely high or low pTau (AT8) pathology is found in donors with and without dementia (*Figure 3A*). Similar cases are found with Aβ pathology in the cortex (*Figure 3B*). These findings are consistent with and extend previous observations in the ACT cohort (*Sonnen et al., 2007*; *Sonnen et al., 2011*) and other community-

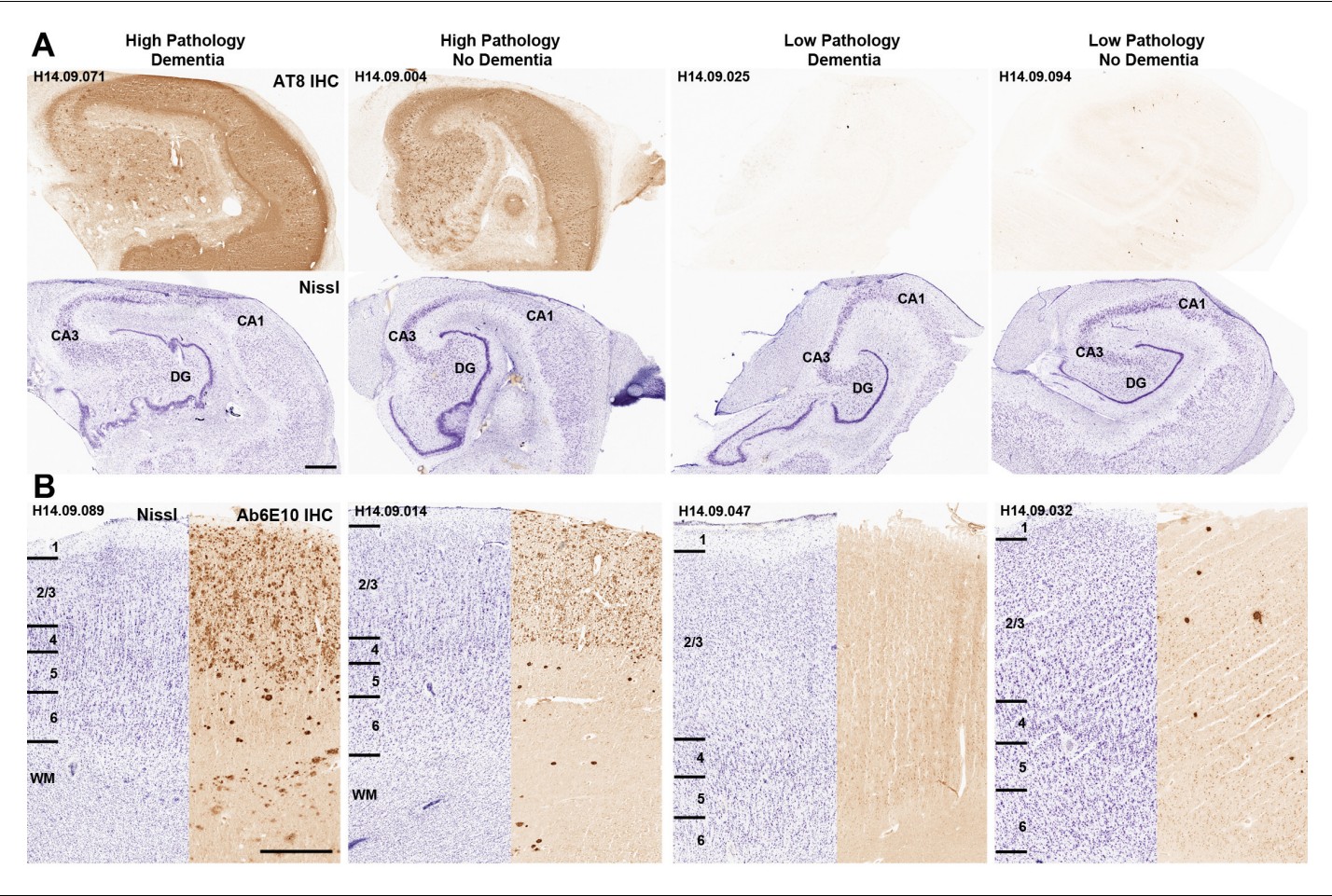

**Figure 3.** Disconnect between pathology and dementia status. AT8 (tau; **A**) and Ab6E10 (Aβ; **B**) IHC in high and low pathology donors with and without dementia, demonstrating individual variation in the relationship between pathology and dementia status. AT8 images of tau pathology are from the hippocampus (with matching Nissl-stained section below), while Ab6E10 images of Aβ pathology are from the parietal cortex. CA1, CA3: hippocampal subfields; DG: dentate gyrus. Numbers in **B** indicate cortical layers. Donor labels are indicated. Scale bar: 1 mm.

DOI: https://doi.org/10.7554/eLife.31126.013

based samples (*Sonnen et al., 2011*), and suggest that some individuals with very AD pathology are resilient to the effects of these pathologies, while others may develop dementia through other mechanisms. Identifying gene expression signatures of resilience and tolerance will be an important area of future study.

## Global and regional molecular signatures of inflammation do not correlate with age or dementia status

Inflammation occurs across a wide range of brain dysfunction, including acute TBI (*Lu et al., 2009*), AD (*Akiyama et al., 2000*) and normal aging (*Franceschi and Campisi, 2014*), due at least in part to disruption of the blood brain barrier (*Popescu et al., 2009*). We took several strategies to assess the range of inflammation across this cohort and the extent to which inflammation is generalized or shows regional specificity, including IHC for microglia (IBA1) and reactive and other types of astrocytes (GFAP), Luminex for cytokines and chemokines, and transcriptome data. Donors showed a continuous range of expression for IBA1 and GFAP, as well as inflammatory proteins based on Luminex assays (*Figure 4A*). A small number of cases (2–5 per region) showed exceptionally high levels, but the majority of cases showed continuous variation across a lower range (high cases excluded from *Figure 4A* to better show the distribution of lower values). Interestingly, individual inflammatory proteins showed regional specificity; for example, some proteins were enriched in hippocampus

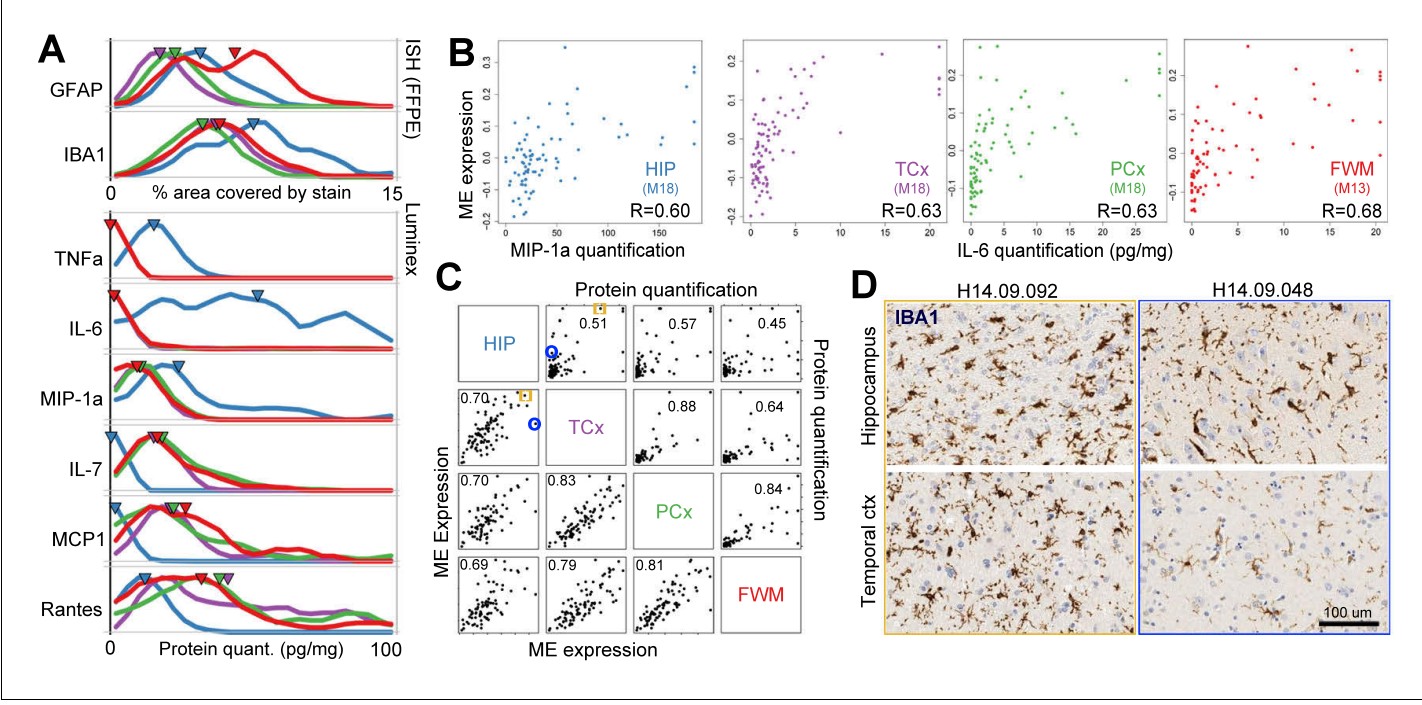

**Figure 4.** Gene expression signatures of inflammation. (a) Distributions of values for glia and for each Luminex variable marking inflammation, separated by brain region (labeling as in *Figure 2A*). (B) Correlation between gene metrics of inflammation (x-axis) and protein metrics of inflammation (y-axis) in each brain region. Gene signatures are defined as the module eigengene (ME) of the module with the largest enrichment for the GO term 'inflammatory response', and protein metrics are the truncated quantifications of the Luminex protein most highly correlated with each ME. (C) Gene (lower left) and protein (upper right) expression markers of inflammation are highly correlated between brain regions. Dots represent donors with x- and y-axes corresponding to the gene and protein values in **B**. Pairwise brain region correlations are shown below in each box. Blue circle and orange box correspond to donors in **D**. (D) IHC for IBA1 in a donor showing inflammation across regions (left, orange), and in a donor showing higher levels of inflammatory marker genes in hippocampus than cortex (right; blue). See also *Figure 4—source datas 1–3*.

DOI: https://doi.org/10.7554/eLife.31126.014

The following source data is available for figure 4:

**Source data 1.** Module assignments and associated module eigengene correlations for each gene in the four regional WGCNA networks.
DOI: https://doi.org/10.7554/eLife.31126.015
**Source data 2.** Module annotation for cell types.
DOI: https://doi.org/10.7554/eLife.31126.016
**Source data 3.** Module comparison with demographic and pathology metrics.
DOI: https://doi.org/10.7554/eLife.31126.017

compared to cortex (TNF-A, IL-6, MIP-1A) or vice versa (MCP1, IL-7, RANTES) (*Figure 4A*). Regional heterogeneity in microglial gene expression has been described elsewhere, such as enrichment in TNF-A expression in rat hippocampus compared to cortex (*Ren et al., 1999*).

Despite this heterogeneity at the individual gene level, there is likely to be a generalized molecular pathway associated with inflammation that can be used to assess the degree of inflammation across regions. Indeed, previous genome-wide transcriptome studies have identified gene networks associated with microglia and inflammation in adult human brain (*Oldham et al., 2008*; *Hawrylycz et al., 2012*; *Miller et al., 2013*; *Zhang et al., 2013*). To identify similar networks in the current cohort we performed weighted gene co-expression network analysis (WGCNA) (*Zhang and Horvath, 2005*; *Langfelder and Horvath, 2008*) separately in each brain region (Materials and methods; see *Figure 4—source data 1* for module assignments). This strategy identifies groups of genes with similar expression patterns in an unbiased manner, whose functional significance can be assigned by searching for overrepresented gene ontology (GO) terms. Here, we identified a network of genes in each region highly overlapping for markers of the GO term 'inflammatory response' (Benjamini and Hochberg corrected $p < 10^{-13}$ in all regions; ToppGene) and for cell-type specific

markers of 'microglia' (Bonferroni corrected $p<10^{-35}$ in all regions) (*Zhang et al., 2014*) or 'Microglial activation' (Bonferroni corrected $p<10^{-12}$ in all regions) (*Mancarci et al., 2016*) (*Figure 4—source data 2*). As expected, these region-specific inflammation gene networks (HIP_M18, TCx_M18, PCx_M18, FWM_M13) show highly significant gene overlap (hypergeometric test, $p<10^{-100}$), despite being generated independently.

Coordinated gene expression levels within gene networks can be summarized by a module eigengene (ME). MEs of these gene networks are highly correlated with specific protein markers of inflammation in specific regions (*Figure 4B*; *Figure 4—source data 3*; $p<10^{-5}$ in all regions; BH-corrected SVA p-values). For example, MIP-1α (gene *CCL3*) shows the best agreement between genes and protein in hippocampus, and IL-6 shows the most consistent patterning in cortex (despite having much higher protein levels in hippocampus). Furthermore, the well-known inflammatory gene *STAT3*, which is activated in mouse brain after induction of inflammatory responses using lipopolysaccharide (LPS) (*Beurel and Jope, 2009*), is one of the genes most highly correlated with the ME in each of these modules (*Figure 4—source data 1*). Using inflammation-related MEs as generalized measures of inflammation, we find high correlations (0.69–0.83; $p<10^{-12}$ for all comparisons) between regions indicating that inflammation is largely a global phenomenon (*Figure 4C*, lower panel). These correlations were significant but lower (0.45–0.88; $p<4\times10^{-4}$ for all comparisons) when correlating individual protein markers across regions (*Figure 4C*, upper panels). We also find some individual cases where inflammation shows regional specificity, both by MEs and using microglial immunohistochemical labeling with IBA1 (*Figure 4D*). This is not unexepcted, as several diseases show region-specific inflammatory responses: for example, the substantia nigra pars compacta is particularly susceptible to neurodegeneration due to inflammation in Parkinson's disease (*Ji et al., 2008*).

None of these inflammation-related gene networks have ME expression significantly correlated with any metrics for aging, cognition, dementia, or associated pathology in this cohort (*Figure 4—source data 3*; p=1 for all comparisons), although a link between inflammation and AD has been described in the literature (*Akiyama et al., 2000*). This discrepancy may be due to the advanced age of this cohort, as there is currently no consensus on the relationship between inflammation and dementia in the oldest old (*Gardner et al., 2013*).

## Transcriptional markers of dementia-related pathology

Gene expression studies have identified dysfunction related to dementia phenotypes in a variety of biological pathways including synaptic transmission, energy metabolism, inflammation, myelin-axon interactions, protein misfolding, and transcription factors (*Colangelo et al., 2002*; *Blalock et al., 2004*; *Webster et al., 2009*; *Miller et al., 2013*), although most of these studies evaluated data from somewhat younger cohorts. In contrast with previous transcriptional studies of AD, we did not find any genes with significant differential expression between control and dementia (or AD) cases in any brain region (SVA, p<0.05, Bonferroni corrected; *Figure 5A* and *Figure 5—figure supplement 1*). We performed several sensitivity analyses that reinforced our conclusion that methodological details were not driving this result (see Materials and methods for details). In addition, none of the gene network MEs described above distinguish dementia cases from controls (*Figure 4—source data 3*).

We extended our analysis to instead search for MEs significantly associated with transcriptional correlates of dementia-related pathologies. No MEs corresponded to any measure of Aβ or α-synuclein pathology, or to age. While the lack of gene expression differences in the brain with age may seem surprising, this result is in line with previous studies that have found a much larger difference in gene expression during middle age (approximately 40–70) than in aged adults (*Lu et al., 2004*; *Berchtold et al., 2008*). However, one gene network (M16, consisting of 660 genes) was significantly correlated with pTau burden in hippocampus (AT8 IHC levels; *Figure 5B*), and showed similar but not statistically significant trends to other pTau metrics (*Figure 4—source data 3*). This set of genes is expressed predominately in astrocytes (Bonferroni corrected $p<10^{-7}$) and microglia ($p<10^{-15}$), potentially marking increased gliosis in hippocampus with increasing tau pathology (*Figure 4—source data 2*); it should be noted that this is not the same gene module associated with inflammation and activated microglia discussed above. Interestingly, several of the hub genes of M16 (genes most highly correlated to the module eigengene) are known to be involved in Aβ processing. For example, ITPKB shows higher expression in human AD than control brain and increases apoptosis

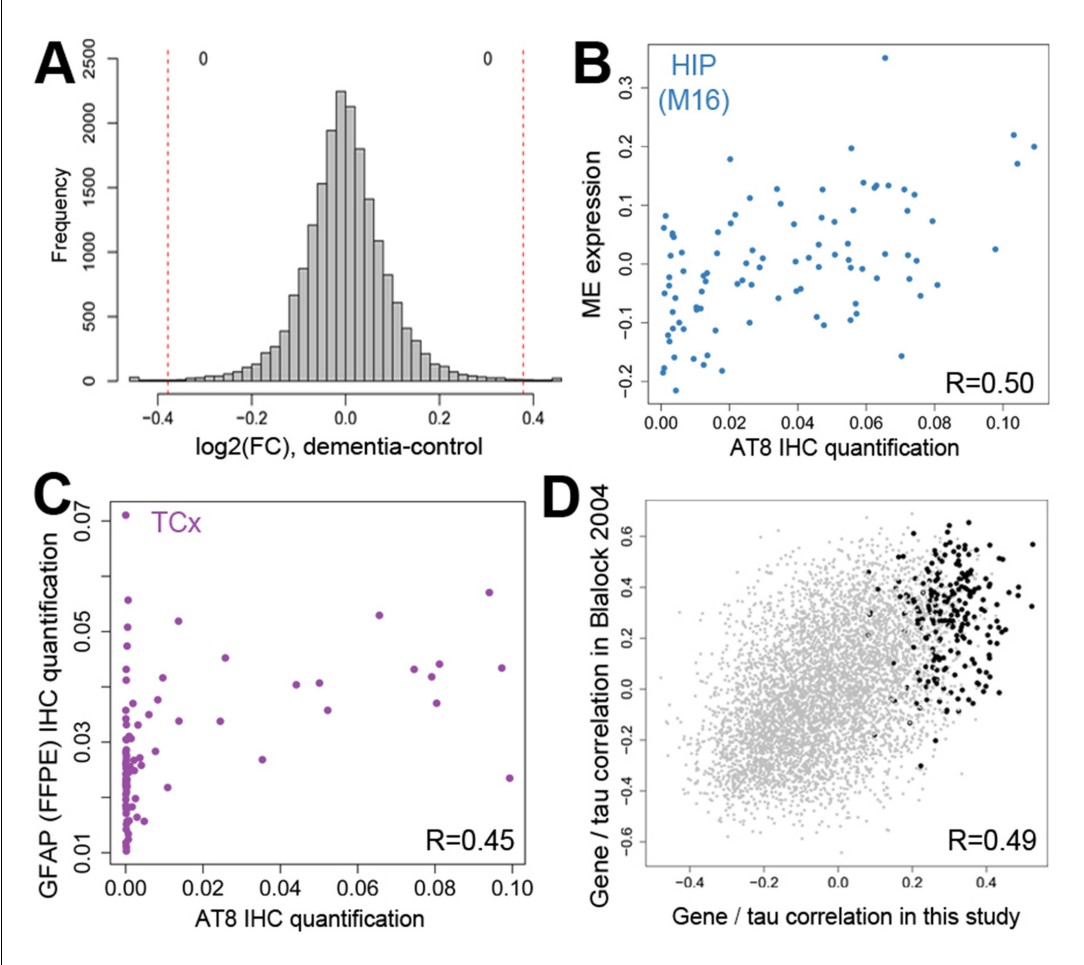

**Figure 5.** Gene expression signatures of dementia and related pathology. (**A**) No significant gene expression differences between donors with and without dementia in hippocampus. The histogram shows the distribution of log₂(fold difference) expression levels (x-axis) between control and dementia donors. Numbers indicate how many genes have a fold change > 1.3 (red lines) and p<0.05. (**B**) Significant correlation between the ME of M16 (y-axis) and measures of tau (AT8 IHC) in hippocampus. (**C**) Significant correlation between protein quantification of IHC for GFAP (y-axis) and measures of pTau (AT8 IHC) in temporal cortex. (**D**) Genes show comparable relationships with tau in this and an earlier study of dementia. X-axis shows the correlation between gene expression and AT8 IHC in this study. Y-axis shows the correlation between quantifications of NFTs and gene expression in (**Blalock et al., 2004**). Dots represent genes, with black dots corresponding to genes in module M16. See also **Figure 5—figure supplement 1**.

DOI: https://doi.org/10.7554/eLife.31126.018

The following figure supplement is available for figure 5:

**Figure supplement 1.** No significant gene expression differences between donors with and without AD diagnosis in RIN-corrected data.
DOI: https://doi.org/10.7554/eLife.31126.019

and Aβ peptide production in mouse Neuro-2a neuroblastoma cells (**Stygelbout et al., 2014**). Similarly, SNX33 increased cleavage of APP alpha-secretase in cultured cells at the cell surface (**Schöbel et al., 2008**), while LRP10 overexpression diverts accumulation of mature APP from the cell surface to the Golgi apparatus, reducing Aβ production (**Brodeur et al., 2012**). Why genes associated with tau metrics would be related to Aβ processing is unclear, but it provides an interesting link between these two pathologies.

Quantifications of IHC for GFAP protein are correlated with pTau burden in temporal cortex, supporting a relationship between tau pathology and reactive astrocytes in this region (**Figure 5C**). Several other modules show significant enrichment for markers of astrocytes, neurons, oligodendrocytes and microglia (**Figure 4—source data 2**), but the MEs of these modules are not associated with

dementia or any related pathology (*Figure 4—source data 3*), in contrast to previously published studies (e.g., [*Miller et al., 2013*]).

To assess whether the gene-pTau trends observed here match prior reports, we calculated the correlation between each gene and AT8 IHC levels, and compared these values with correlations between gene expression and reported levels of pTau in the hippocampal CA1 region (*Blalock et al., 2004*). The two studies agree well, with a correlation over all genes of R = 0.49 (*Figure 5D*, genes in M16 in black). Together, these results recapitulate the reported relationship between astrocyte and microglia-related gene expression and pTau pathology in hippocampus, but fail to identify genes related to dementia status.

## Dementia-related gene expression associated with variation in RNA quality

Our failure to identify genes significantly related to dementia status was surprising, given that many studies have shown differential gene expression with AD (*Colangelo et al., 2002*; *Blalock et al., 2004*; *Liang et al., 2008*; *Webster et al., 2009*; *Avramopoulos et al., 2011*; *Miller et al., 2013*; *Zhang et al., 2013*; *De Jager et al., 2014*; *Satoh et al., 2014*; *Allen et al., 2016*). However, we found an inconsistency in how these studies normalized for tissue quality as measured by the pH or RIN scores of the tissues analyzed; in fact, only a few of them corrected for RIN at all. Repeating our analysis without accounting for RIN, we find a large fraction of genes (11%) to be differentially expressed between dementia cases and controls in at least one region (B & H corrected p<0.05, $\log_2$(FC) >1.3; *Figure 6—source data 1*), leading to apparently larger fold changes between conditions (*Figure 6A*; compare with *Figure 5A*). We once again find very few genes significantly associated with dementia when we perform this analysis on a subset of 70 donors in our cohort matched for RIN, sex, and dementia status (Materials and methods; two or fewer in each region). This result suggested a direct link between RNA quality and dementia status. Indeed, we find a substantially lower RNA quality in dementia cases vs. controls in all four brain regions (*Figure 6B*; *Figure 6—figure supplement 1B*), and this difference was not related to the time between death and autopsy (PMI <8 hr, all donors).

We next repeated this comparison on data from four additional population-based cohorts as part of the AMP-AD knowledge portal (https://www.synapse.org/ampad; (*Bennett et al., 2012a*; *Bennett et al., 2012b*; *Allen et al., 2016*)), and compared these with previous reports (*Colangelo et al., 2002*; *Preece and Cairns, 2003*; *Durrenberger et al., 2010*; *Zhang et al., 2014*). In five of the eight additional data sets assayed, donors with AD had significantly lower RIN than donors without dementia (*Table 2*). Donors with AD had significantly higher RNA quality in only one study (*Allen et al., 2016*). Thus the link between dementia status and RNA-quality is a broader phenomenon that is not unique to the ACT cohort, but is also not ubiquitous.

To assess the impact of RNA quality on gene expression levels, we compared gene expression to RIN. We find that 47% of expressed genes are correlated with RIN (Benjamini and Hochberg corrected p<0.05, RIN correlation >0.5; *Figure 6C*; *Figure 6—source data 2*)—in some cases accounting for 80% of the variation in gene expression—and that failure to account for RNA quality reduces our ability to separate samples by brain region (*Figure 6—figure supplement 2*). This result mirrors other studies that have shown RNA quality dramatically impacts measured gene expression levels (*Preece and Cairns, 2003*; *Li et al., 2004*; *Tomita et al., 2004*; *Mexal et al., 2006*; *Vawter et al., 2006*; *Atz et al., 2007*). However, we extend this result to show that the set of genes positively correlated with RIN are enriched for pathways found to be disrupted with AD in prior gene expression studies (*Colangelo et al., 2002*; *Blalock et al., 2004*; *Webster et al., 2009*; *Miller et al., 2013*), including the GO terms 'mitochondrion organization' and 'RNA processing' (Benjamini and Hochberg corrected $p<10^{-30}$; *Figure 6—source data 3*). But is the entire relationship of genes to dementia a function of RIN? To test this, we rank ordered genes by fold change difference between controls and dementia cases with and without correcting for RIN. In all regions, we found a significant correlation (R ~ 0.8) between RIN-corrected and uncorrected rankings (*Figure 6D*), indicating that much of the variance is not explained by RIN.

These results suggest that differential gene expression in neurodegeneration may include both the contributions of chronic conditions (e.g., dementia) and acute conditions (e.g., agonal stress), and/or other factors impacting RNA quality. To test whether previously published studies of AD could show similar effects, we compared gene measures of RNA quality (RIN) and of dementia status

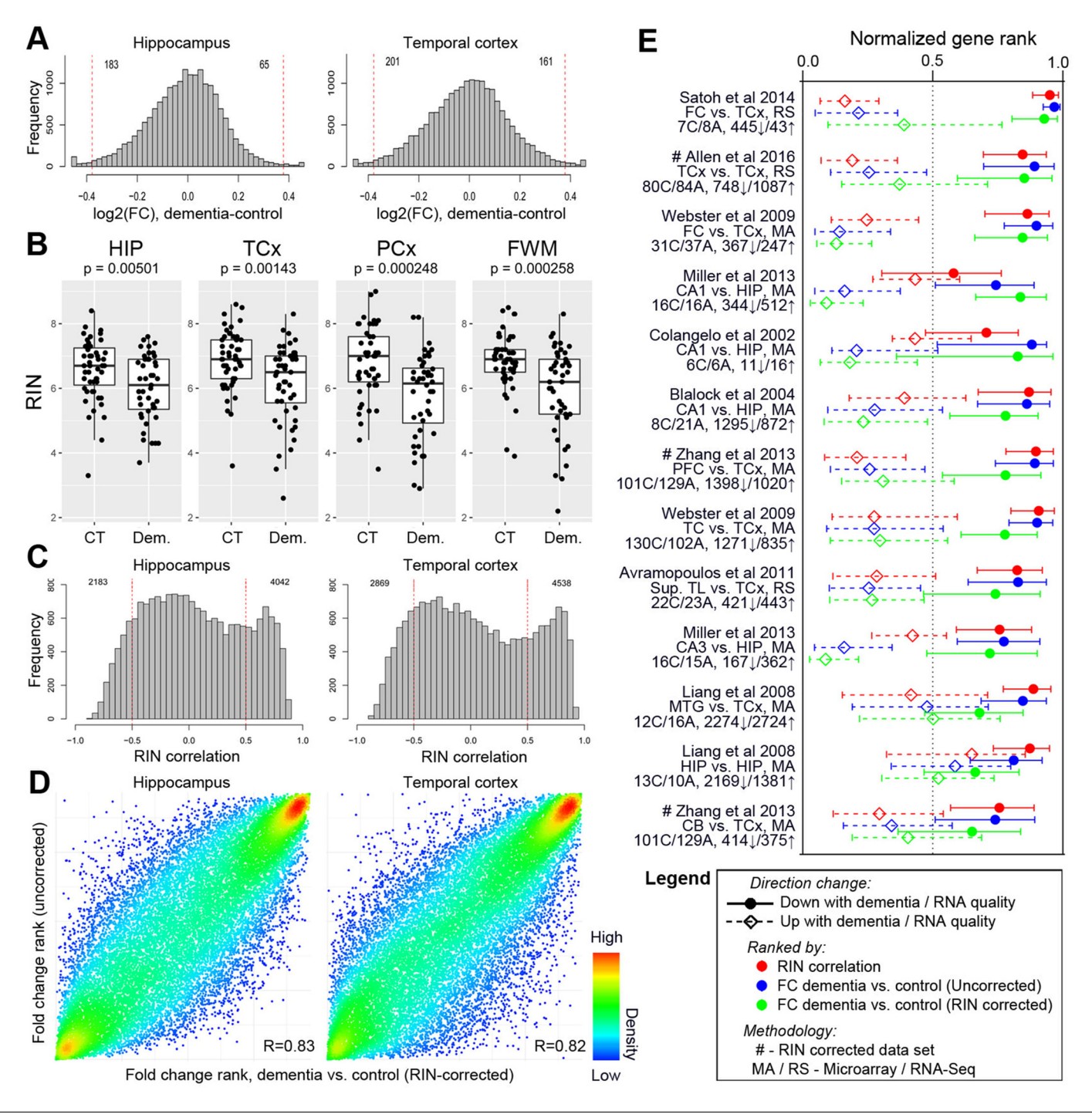

**Figure 6.** Differences in RNA quality between dementia and controls greatly impact gene expression results. (**A**) Gene expression differences between donors with and without dementia in uncorrected data. Histograms show the distribution of log$_2$(fold difference) expression levels (x-axis) between control and dementia donors in two brain regions (hippocampus, left; temporal cortex, right). Numbers indicate how many genes have a fold change > 1.3 (red lines) and p<0.05. (**B**) RNA quality in donors with dementia (Dem.; right bars) is significantly lower than in non-demented controls (CT; left bars) in all four brain regions. Y-axes are RIN values. Plots as in *Figure 2B–D*. (**C**) Gene expression levels for many genes are highly correlated with RIN, with more showing lower expression with lower RNA quality (positive values) than with higher RNA quality (negative values). Histograms show the distribution of RIN correlations in two brain regions. Numbers indicate how many genes have R > 0.5 (red lines) and p<0.05. (**D**) Rank order of fold differences between controls and dementia cases is largely unchanged after controlling for RNA quality. Ranked fold differences on the x- and y-axes correspond to *Figure 6A* and *Figure 5A*, respectively. Dots indicate genes and are color-coded by density. (**E**) Genes with higher or lower expression levels in people with dementia compared with cognitively normal older adults from 12 brain regions in eight previous studies (rows) are related to

*Figure 6 continued on next page*

*Figure 6 continued*

dementia diagnosis and RNA quality in this study. Horizontal tics show the 25th percentile, median, and 75th percentile rank of the indicated dementia-related list in our current data set. Gene expression levels from genes lower in low RIN samples are also lower in AD samples from the comparison studies (red, solid lines are shifted towards 1), while gene expression levels from genes higher in high RIN samples are also higher in AD samples from the comparison studies (red, dotted lines are shifted towards 0). Gene expression results accounting for RIN (green) generally agree less well between studies than results not accounting for RIN (blue). See also *Figure 6—figure supplements 1–2* and *Figure 6—source datas 1–6*.
DOI: https://doi.org/10.7554/eLife.31126.020

The following source data and figure supplements are available for figure 6:

**Source data 1.** Log2 fold changes between AD vs. control and dementia vs. control for each gene in all four brain regions, along with associated SVA p-values.
DOI: https://doi.org/10.7554/eLife.31126.023

**Source data 2.** Correlations between each gene and RNA quality (RIN) with associated SVA p-values in all four brain regions.
DOI: https://doi.org/10.7554/eLife.31126.024

**Source data 3.** Significant GO terms for genes increasing or decreasing expression with decreasing RNA quality.
DOI: https://doi.org/10.7554/eLife.31126.025

**Source data 4.** Description of nine previous studies comparing AD vs. control, including details of how gene lists used in paper were derived.
DOI: https://doi.org/10.7554/eLife.31126.026

**Source data 5.** List of 26 gene lists from the nine above publications that are used in *Figure 6E*.
DOI: https://doi.org/10.7554/eLife.31126.027

**Source data 6.** Significance of association between each of the first 25 principal components and every assessed metric in the un-normalized and RIN-normalized data.
DOI: https://doi.org/10.7554/eLife.31126.028

**Figure supplement 1.** Differences in RNA quality between AD and controls greatly impact gene expression results.
DOI: https://doi.org/10.7554/eLife.31126.021

**Figure supplement 2.** Decreased separation of brain region in principal component space without RIN correction.
DOI: https://doi.org/10.7554/eLife.31126.022

(before and after accounting for RIN) in our data set with AD-related genes from nine previous studies (*Figure 6—source datas 4–5*) (*Colangelo et al., 2002*; *Blalock et al., 2004*; *Liang et al., 2008*; *Webster et al., 2009*; *Avramopoulos et al., 2011*; *Miller et al., 2013*; *Zhang et al., 2013*; *Satoh et al., 2014*; *Allen et al., 2016*). First and foremost, we found that many of the genes most highly associated with dementia status were shared between studies, whether we controlled for RIN or not (*Figure 6E*, green and blue), confirming previous reports. However, genes that had lower levels of expression among people with AD tended to also have lower expression in donors with low

**Table 2.** RNA quality assessment for donors with and without AD in multiple studies.

| Data set | Brain region | Count (Control\|AD) | RIN/pH in control | RIN/pH in AD | P-value |
|---|---|---|---|---|---|
| ACT cohort (current data set) | TCx | 50\|29 | 6.87 ± 0.92 | 6.18 ± 1.27 | p=1.48×10⁻² |
| ROS (*Bennett et al., 2012a*) | TCx | 107\|136 | 7.21 ± 1.01 | 7.06 ± 0.95 | p=0.23 (ns) |
| MAP (*Bennett et al., 2012b*) | TCx | 94\|120 | 7.25 ± 1.07 | 6.79 ± 0.96 | p=1.18×10⁻³ |
| MSBB * | BA36 | 98\|169 | 6.51 ± 1.30 | 5.86 ± 1.59 | p=1.10×10⁻³ |
| Mayo Study (*Allen et al., 2016*) [†] | TCx | 31\|82 | 7.64 ± 1.21 | 8.59 ± 0.55 | p=1.98×10⁻⁴ |
| (*Colangelo et al., 2002*) | CA1 | 6\|6 | 6.75 ± 0.1 | 6.76 ± 0.1 | ns |
| (*Preece and Cairns, 2003*) [$] | Cortex | 81\|90 | ~6.5 | ~6.4 | p<10⁻² |
| (*Durrenberger et al., 2010*) [‡,$] | Brain | 72\|12 | ~7.1 | ~5.9 | p<10⁻⁴ |
| (*Zhang et al., 2014*) | PFc | 101\|129 | 7.31 ± 0.47 6.55 ± 0.34 | 7.12 ± 0.56 6.37 ± 0.32 | p=6.78×10⁻³ p=1.22×10⁻⁴ |

*=we are defining AD as CDR >1 in this data set, but note that the result holds for other cutoffs, †=only control donors from the Mayo Brain Bank Dickson were considered, ‡=Tissue collected from multiple brain regions and multiple brain banks. RNA quality is assessed with either RIN (black text) or pH (blue text) in each study. $=RIN/pH values are estimated from plots. Calculated p-values in this table are two tailed student t-tests uncorrected for multiple comparisons, and p-values from previous studies are as reported.
DOI: https://doi.org/10.7554/eLife.31126.029

RNA quality (*Figure 6E*, red), in some cases agreeing better with RIN reported herein than with dementia status reported herein. The converse was also true, although the effect was less robust. Furthermore, in nearly every case, our dementia-related genes identified without accounting for RIN agree better with gene lists from previous studies than those identified when we did account for RIN. The same results hold when we repeat this entire analysis considering only the subset of dementia cases with AD (*Figure 6—figure supplement 1*). These results, which were consistent across studies implementing a wide variety of experimental designs and strategies for controlling for RNA quality (or not), demonstrate a strong relationship between transcriptional changes in neurodegenerative disease and RNA quality, although it is important to note that at least two of these studies (*Zhang et al., 2013*; *Allen et al., 2016*) identified several hundred dementia-related genes after controlling for RIN, indicating that relative contribution of neurodegenerative disease and RNA quality to gene expression may differ between studies. Finally, we note that all our results hold when considering the top 25 principal components instead of differential genes or WGCNA modules (*Figure 6—source data 6*). For example, exactly one PC (PC1 or PC2 for all regions) is significantly associated with dementia status in the unnormalized data, while none are associated with dementia in the RIN-normalized data.

## Discussion

The Aging, Dementia and TBI Study aims to provide the research community with an open access multimodal resource for studying relationships between cognition, neurodegeneration, inflammation, and injury in the aged brain (http://aging.brain-map.org/). This resource includes gene expression, protein quantification of neuropathology and inflammatory molecules, and histology on markers for cell type and neuropathology in four brain regions from 107 well-characterized donors from the ACT cohort. This unusually broad and systematic study allows a variety of analyses to correlate these features and identify associated molecular pathways. We confirm known associations between pTau and Aβ pathologies and dementia, and identify sets of co-expressed genes correlated with tau pathology and inflammation. The advanced age of the ACT cohort presented much higher variability than is seen in somewhat younger cohorts, which may have led to our difficulties in identifying significant gene signatures associated with aging, dementia and neuropathological markers, since these relationships appear to be stronger in younger individuals. Furthermore, we confirm a systemic relationship between dementia and RNA quality, showing a strong overlap between genes whose levels are affected by RNA quality and genes previously reported to be associated with AD. This study illustrates the challenges posed by the high degree of biological variation in the aged brain, and provides a resource that will facilitate future efforts to understand the nature of this variation.

We find a relationship between pTau (and to a lesser extent Aβ) neuropathology and cognitive status, based both on global pathology metrics (e.g. Braak score) and local hippocampal and cortical quantification of pathological tau (e.g., AT8 IHC), even in patients older than 90 years old. Interestingly, the strongest correlations between tau pathology and cognitive scores, dementia status and gene expression were observed with pTau protein quantification based on automatic image analysis of IHC data. This strategy for image quantification, which has been successfully applied at a large scale to study gene expression patterns in mouse (*Lein et al., 2007*), has the potential to provide an unbiased and informative method for studying pathology. A prior study of 390 donors also found a strong relationship between antemortem global cognitive ability (MMSE score) and counts of NFTs, and a weaker correlation with neuritic plaques (*Nelson et al., 2007*). However, the connection between neuropathology and cognitive status was weaker in the current cohort and there were many cases with a striking disconnect. For example, 25% of donors with Braak stage >= 5 did not have dementia. A major reason for the weaker relationship between pTau pathology and cognitive status appears to be higher levels of tau pathology associated with advanced age that is not related to dementia (*Haroutunian et al., 2008*; *Corrada et al., 2012*). This finding could be related to primary age-related tauopathy (PART), a recently described age-associated pathologic entity, usually in non-demented individuals, characterized by pTau-related neurofibrillary degeneration in the absence or paucity of Aβ pathology (*Crary et al., 2014*). As a new entity, PART has not been widely assessed in the ACT study autopsy cohort, and the co-existence of PART and AD, particularly in the oldest old, is poorly characterized, although there is some evidence of co-occurrence of these diseases

(*Mungas et al., 2014*). The influence of mixed pathologies on cognition is poorly understood and an area of active investigation. The relationship between tau pathology and dementia status in our study was not observed in donors > 90 years old, highlighting the complexities of neuropathological processes in the oldest old and demonstrating resilience to the effects of pathology ('tolerance') in a substantial subset of aged individuals.

Neuroinflammation, predominantly in the form of innate immune activation via microglia, has been reported to occur across normal aging and in many neurodegenerative disorders, including AD (*Akiyama et al., 2000*; *Franceschi and Campisi, 2014*). Here, we took advantage of the wide range of molecular analyses to search for such correlations in the aged brain. A challenge we found is that individual protein markers of inflammatory pathways behave differently in different structures and differently from each other (*Ren et al., 1999*; *Ji et al., 2008*; *Wang et al., 2015*). To identify more global signatures of inflammatory pathways we used gene network analysis to identify sets of co-expressed genes enriched for inflammatory genes. This approach identified inflammatory gene networks in each brain region that were correlated with the most informative individual protein markers in those regions. The gene expression levels of these networks were highly correlated across brain regions, indicating that neuroinflammation is in many cases largely a global phenomenon across brain regions, although there were individuals with regional enrichment of inflammatory signatures. Neither the individual inflammatory markers nor gene expression levels of the inflammatory gene network correlated with either aging or dementia status.

We identified important variation in the RNA quality of tissues analyzed. Surprisingly, we found a statistically significant relationship between RNA quality and dementia status in donors from several independent cohorts. Why might RNA quality vary with dementia status? One possibility is that RNA quality reflects antemortem conditions, and control donors are more likely to expire from sudden, unexpected causes than donors with neurodegenerative conditions requiring long-term care (*Monoranu et al., 2009*; *Mills et al., 2014*). A number of studies have shown that gene expression levels can vary dramatically based on end-of-life conditions (*Li et al., 2004*; *Tomita et al., 2004*; *Atz et al., 2007*; *Monoranu et al., 2009*; *Durrenberger et al., 2010*). For example, Li and colleagues used unbiased clustering of brain tissue from multiple regions to group donors into two types differing by brain pH and agonal duration (*Li et al., 2004*). Remarkably, 30–50% of all genes differentiated these two types, including many markers for oxidative stress which have lower expression in donors with lower RNA quality (*Vawter et al., 2006*). We find similar results: 47% of expressed genes are correlated with RIN, including many in mitochondrial-related pathways whose levels decrease with RIN, although information regarding end of life conditions is unavailable for ACT cohort donors.

Relationships between gene expression, RNA quality, and agonal state markedly complicate studies of neurodegenerative diseases (*Monoranu et al., 2009*; *Mills et al., 2014*). Here we find a strong overlap between signatures of AD and RNA quality, with most (but not all) of the transcriptional variability accounted for by RIN rather than disease status. To the best of our knowledge, this is the first study directly comparing genes associated with RNA quality and dementia status. Furthermore, we find that nearly all reported AD gene lists (regardless of treatment for RNA quality) are strongly enriched for gene expression in genes associated with both RIN and AD status in this study, and that accounting for RNA quality in this study markedly decreases the agreement between studies. Importantly, all previous studies included in our analysis that did control for RNA-quality identified many dementia-associated genes, suggesting that the relative impact of RNA-quality in our data set may be exaggerated. Nevertheless, these results suggest a potential convergence of gene pathways involved in agonal state and dementia, and highlight the importance of carefully accounting for technical variables—particularly RNA quality—when studying neurodegenerative diseases.

## Materials and methods

### Participant information and consent

All work was performed according to guidelines for the research use of human brain tissue. Participants signed informed consent forms at enrollment that includes permission for sharing de-identified data, and signed additional consent forms for the autopsy that included data and tissue sharing. Autopsy consents were updated for all subjects with the legal next of kin after death. All study

procedures were reviewed and approved by Institutional Review Boards at Kaiser Permanente Washington and the University of Washington. Non-identifying information about each 107 participants (i. e., age, sex, etc.) is publicly available under the 'Specimens' tab at http://aging.brain-map.org/.

## ACT cohort

ACT is a prospective, longitudinal study of randomly-selected, cognitively normal participants of Kaiser Permanente Washington in the Seattle area that were willing to volunteer for the study (*Kukull et al., 2002*; *Larson et al., 2006*; *Crane et al., 2013*). Kaiser Permanente Washington is an integrated staff-model Health Maintenance Organization (HMO). At enrollment and at follow-up visits every two years, ACT research staff members administer the Cognitive Abilities Screening Instrument (CASI) (*Teng et al., 1999*) and participants with CASI ≤85 receive secondary follow-up with a clinical evaluation and a comprehensive neuropsychological battery. Results of these evaluations and clinical data are reviewed in a multidisciplinary consensus conference which uses standardized criteria to diagnose incident dementia (*American Psychiatric Association, 1994*) and AD as well as other neurodegenerative diseases when applicable (*McKhann et al., 1984*). Total enrollment as of December 2015 was approximately 5100 people, including more than 600 participants who have donated their brains. Requests to access other data from the ACT cohort should be addressed to KPWA.act-proposals@kp.org.

## Tissue collection and utilization

A team from the University of Washington (UW) Neuropathology (NP) Core is contacted soon after death to perform brain autopsies of ACT subjects. ACT study staff ask participants at enrollment and every study visit whether they have experienced a loss of consciousness (LOC) and, if so, what caused it, such as electrocution, near drowning, or head injury (TBI). If updated autopsy consent is obtained and the brain can be removed and dissected with a postmortem interval (PMI) <8 hr, a rapid autopsy is performed. During a rapid autopsy, ventricular cerebrospinal fluid (CSF) is taken, the brain is hemisected along the mid-sagittal plane and dissected and ~60 flash frozen tissue samples from at least 12 brain regions are collected, flash-frozen in liquid nitrogen, and stored at −80℃. The unsampled hemibrain, and all remaining non-frozen tissue from the contralateral hemisphere is then fixed in 10% normal buffered formalin for approximately 2–3 weeks. Fixed tissue from every brain undergoes a thorough neuropathological examination where 22 standard samples, in addition to samples of any focal lesions or abnormalities, are dissected and submitted for routine processing for formalin-fixed paraffin-embedded (FFPE) sections.

This project was initially designed to study the long-term effects of TBI exposure, and participants were selected on the basis of exposure or lack of exposure to TBI. All ACT subjects with a TBI with loss of consciousness (LOC) and rapid autopsy with available banked frozen tissue were identified, and then each TBI with LOC donor was matched for sex, age, year of death, and finally PMI to an individual in the ACT autopsy sample without a history of TBI with LOC. Once a subject was included in this study, two adjacent flash frozen tissue blocks from parietal lobe, temporal lobe, and hippocampus were removed from the NP Core repository; one was sent to the Allen Institute for IHC, ISH and RNA -seq, and the other processed at the University of Washington (UW) for immunoassays (Luminex) and gas chromatography-mass spectrometry (GC/MS). If two blocks were not available, the remaining block was either divided (cortex) or prioritized for Allen Institute studies (hippocampus).

## Tissue processing for histology and immunoassays

Frozen tissue at UW was divided evenly (while frozen, in the sagittal plane through the long axis of the gyrus) for GC/MS (isoprostanes), where the entire piece was used, and for immunoassays (Luminex), which was run on 1 cm punch biopsies from the depth of sulcus cortex (gray matter) in parietal and temporal lobe and through deepest subcortical white matter (in the parietal lobe sample). Due to relative paucity of available tissue, the entire hippocampus tissue block was submitted for immunoassays. For IHC of FFPE tissues, blocks were taken from either the same (cortex) or the opposite (hippocampus) side of the brain that was sampled for frozen tissues (although not from adjacent blocks), and were submitted for sectioning, histochemical, and immunohistochemical staining. Slides were then sent to the Allen Institute for scanning and image analysis as described below. Frozen

tissue sent to the Allen Institute was cryosectioned into a series of 25-micron-thick sections that were designated for histological staining (Nissl, ISH, IHC, and Thioflavin-S) and for RNA-seq (see below). Following sectioning, histological stains were processed according to standard protocols as previously described (*Sun et al., 2002*; *Lein et al., 2007*). GC/MS was quantified as described previously (*Montine et al., 2005*). Specific assays run for GC/MS, Luminex, ICH, and ISH are presented in *Figure 1*. IHC markers for paired helical filament pTau (AT8) and Aβ (Ab6E10) were processed on both fresh frozen and FFPE tissue. Note that a broader marker for pTau, Tau2, was processed only for FFPE tissue (see *Figure 1—figure supplement 1* for an example staining).

## Image processing and quantification

Nissl, H&E-LFB, ISH and IHC slides were scanned at 10x full resolution using a Leica ScanScope scanner, while Thioflavin-S slides were scanned at 10x full resolution using an Olympus VS110 scanner. An Informatics Data Pipeline (IDP) managed image preprocessing, image QC, IHC expression detection and measurement, Nissl processing, annotation QC and public display of information via the web application, as described previously (*Dang et al., 2007*), with some modifications and additions for processing images for this project. For ISH and IHC slides, respectively, masks highlighting areas with enriched gene expression or immunoreactivity were generated using adaptive detection/segmentation image processing algorithms. Images that were out of focus after rescanning or with technical or tissue artifacts obscuring the target anatomical region were then failed and excluded from public release. For each set of gene images available in the online viewer, the nearest set of Nissl-stained sections (and other histological data) can be accessed and viewed. To generate quantitative image metrics for IHC, macrodissection sites as delineated on the Nissl images were used to annotate regions of interest (ROIs) on each of the near-adjacent IHC images. The ROI was then adjusted if there were technical artifacts that would affect the evaluation of pathology. The expression density, defined as the percentage of area within the ROI that was occupied by the IHC reaction product, was then assessed using an adaptive detection/segmentation technique which algorithmically determines whether each pixel in an ROI contains the IHC stain (see the Expression Detection Module section in the Informatics Data Processing paper in the Allen Mouse Brain Atlas Documentation tab for more details; http://mouse.brain-map.org/static/docs). For stains with very low expression densities, ROIs that were identified as outliers were visually inspected and then adjusted or excluded as necessary. Good correlations were seen between quantifications of antibodies for amyloid beta and pTau in FFPE and fresh frozen tissue, indicating good agreement between these two measures of pathology (*Figure 2—figure supplement 2*).

## RNA-Seq tissue and RNA processing

Collection of tissue samples from temporal and parietal neocortex, parietal white matter, and hippocampus was done by manual macrodissection. Specific areas for macrodissection were identified by neuroanatomists using images of Nissl-stained tissue sections immediately adjacent to the sampled tissue, and were excised from the remaining tissue frozen tissue block using a scalpel. Tissue was immediately transferred to prepared tubes where RNA was isolated using the RNeasy Lipid Tissue Mini Kit (Qiagen #78404) as per manufacturer's instructions. RNA was then quantified on a Nano-drop 8000 spectrophotometer (Thermo Scientific, Wilmington, DE) and normalized to 5 ng/μl before RNA QC was performed using a Bioanalyzer (Agilent Technologies) and RNA Integrity Number (RIN) was recorded. Total RNA (250 ng) was used as input into the Illumina TruSeq Stranded Total RNA Sample Prep Kit (RS-122–2203), which uses random hexamer first strand cDNA synthesis and includes rRNA depletion (Ribo-Zero Gold rRNA depletion kit to remove both cytoplasmic and mitochondrial rRNA) and fragmentation. At the time of project inception, this sequencing strategy provided the most reliable option for quantification of transcriptomic reads from tissue of widely varying quality, allowing the broadest inclusion of donors from the ACT cohort. External RNA Controls Consortium (ERCCs) (*Baker et al., 2005*) at a 1:10,000 dilution were spiked into each sample. RNA sequencing was done on Illumina HighSeq 2500 using v4 chemistry, producing a minimum of 30M 50 bp paired-end clusters per sample. Expression Analysis, Inc. (Morrisville, NC) performed both the TruSeq Stranded Sample Prep as well as the Illumina sequencing. All samples, regardless of RNA quality, were sent for sequencing. In total 377 samples from 107 donors passed all QC metrics and are included as part of the resource. Nearly all of the missing 51 samples were excluded

because tissue was unavailable from the brain bank or because it completely failed in sequencing. A few samples were failed because their average inter-array correlation across all genes was several (usually but not always 3) standard deviations below the mean of all other samples from the same brain region. This strategy has been used to fail samples in other Allen Brain Atlases and is useful for ensuring that results are not driven by outliers. For this analysis, we removed one additional sample from the data set that showed high expression of Y chromosome genes but that was collected from a genetically-confirmed female who was documented to have previously given birth (donor H14.09.011).

## RNA-Seq data alignment and normalization

Raw read (fastq) files were aligned to the GRCh38.p2 human genome (current as of 01/15/2016). Illumina sequencing adapters were then clipped using the fastqMCF program (*Aronesty, 2011*), and then mapped to the transcriptome using RNA-Seq by Expectation-Maximization (RSEM) (*Li et al., 2010*) using default settings except for two mismatch parameters: bowtie-e (set to 500) and bowtie-m (set to 100). RSEM aligns reads to known isoforms and then calculates gene expression as the sum of isoform expression for a given gene, assigning ambiguous reads to multiple isoforms using a maximum likelihood statistical model. Reads that did not map to the transcriptome were then aligned to the hg38 genome sequence using Bowtie with default settings (*Langmead et al., 2009*), after which remaining unmapped reads were mapped to ERCCs. Anonymized BAM files (where sequence-level information has been removed) for both transcriptome- and genome-mapped reads, and gene-level quantification (transcripts per million (TPM), fragments per kilobase per million (FPKM), and number of reads) are available as part of the resource (see Download tab).

For analysis, the FPKM data matrix was first adjusted for the total transcript count using TbT normalization (*Kadota et al., 2012*), which scales each sample based on the summed expression of all genes that are not differentially expressed. The differential expression vector was defined as TRUE if a sample was from either temporal or parietal cortex, and FALSE otherwise. Sample data were then log-transformed and scaled such that the total $\log_2$(FPKM + 1) across the entire data set remained unchanged after normalization. The result was that all expression levels for a particular sample were multiplied by a scalar close to 1 (in most cases between 0.9–1.2).

The amount of variation explained by each demographic and tissue source was estimated using MDMR (*Zapala and Schork, 2012*). Specifically, Pearson correlation-based distances were calculated between each pair of samples ($D_{xy}$ = 1-corr(x,y)), and a matrix of these values and of each demographic and tissue source variable was input into the MDMR R function as a univariate model with 100 permutations. Resulting percent variance explained and associated p-values are presented in *Figure 1—figure supplement 2*.

As brain region and RIN were identified as the largest sources of variability, log-normalized quantifications of each gene were corrected for RNA quality independently within each brain region. This was done as follows: (1) exclude outliers (>3 standard deviations from the mean) and zero values, (2) determine whether expression data is best fit by one or two Gaussians using Mclust (*Fraley and Raftery, 2002*), (3) model RIN as a quadratic variable in each of the one or two groups, taking the sum of the residual and the mean as the normalized value, and then setting any negative values to 0. In most cases this normalization is equivalent to correcting the log-transformed data for RIN + adjusted RIN, and in other cases can additionally account for bimodalities in the data (e.g., gender) that are unrelated to RNA quality. This strategy of regressing out RIN is conceptually similar to one previously published (*Gallego Romero et al., 2014*). Other strategies accounted for RNA quality in RNA-seq data assume cDNA synthesis based on poly-A priming (*Wan et al., 2012*; *Sigurgeirsson et al., 2014*); these models break down in our data set where cDNA synthesis is based on a random hexamer method.

We note that this is a different final normalization step from that performed in the online data resource, where data were corrected for RIN + adjusted RIN + batch in linear space. The current strategy of excluding outliers from the normalization retains realistic expression values for biologically-relevant processes such as inflammation, and accounting for bimodalities removes effects of sex (which we sought to retain on the web resource). Similarly, batch correction is not included in this analysis as donors with the most severe TBIs were front loaded in the first two batches due to experimental constraints. Using data normalized on the website to assess differential gene expression between donors with and without dementia (as described below) produced comparable results.

## Assessment of differential and co-expression

For pathology and demographic information, significance of differential expression between groups was assessed with analysis of variance (ANOVA) tests using the 'aov' function in R (*Chambers et al., 1992*). Correlations between continuous variables were calculated using the 'cor' function in R and are Pearson correlations with Bonferroni-corrected p-values of p<0.05, unless otherwise specified. Distributions of quantitative metrics are displayed using a smoothed density curve, with no associated statistical tests performed. Two tailed student t-tests were used to compare RIN between control and AD donors from multiple studies. We used surrogate variable analysis (SVA) (*Leek and Storey, 2007*) to quantify significance of gene expression with respect to dementia status (in combination with fold-difference thresholds) and RNA quality (in combination with correlation thresholds). P-values of p<0.05, after Benjamini and Hochberg correction, were considered significant unless otherwise noted. SVA was also used for assessing significance in gene clusters, as discussed below.

To assess the robustness of our result that few if any genes are significantly associated with dementia (or AD) status after controlling for RIN, we performed additional analyses to quantify significance of gene expression with respect to dementia status, in all cases defining significance as p<0.05 after Bonferroni correction. First, we repeated our analyses using additional statistical tests including 1) two tailed student t-tests, 2) ANOVA, 3) and limma (*Ritchie et al., 2015*), in all cases defining two groups based on dementia (or AD) status. Second, we repeated the SVA analysis described above on the RIN-normalized RNA-Seq data available for download from the website (which uses a slightly different normalization schema, as described above). Third, we performed principal component analysis (PCA) independently on each region using all genes, and used SVA to assess whether any of the top 25 PCs showed significant associate with dementia. Finally, we sub-sampled our data set to 70 donors who are matched for RIN, sex, and dementia status and repeated the SVA analysis using data that is not RIN-corrected to determine whether our particular RIN-normalization strategy could be biasing our ability to identify genes associated with dementia. In all cases we found two or fewer total genes associated with dementia or AD, indicating that our negative result is not due to improper statistical assessment.

We used weighted gene co-expression network analysis (WGCNA) (*Zhang and Horvath, 2005*; *Langfelder and Horvath, 2008*) to generate unbiased gene co-expression networks separately for each brain region. Initial networks were generated using an automated strategy with the following function call in R: blockRun = blockwiseModules(datExprRun, checkMissingData = TRUE, maxBlockSize = 17500, power = 14, networkType = 'signed', deepSplit = 2, minModuleSize = 20, minCoreKMESize = 7, minKMEtoStay = 0.4, mergeCutHeight = 0.1, numericLabels = TRUE, verbose = 1) where datExprRun is the $\log_2$ normalized RNA-seq data from the top 9615 (50%) most variable genes (in each region). We then calculated module eigengenes (ME), defined as the first principal component of genes in the module. If the resulting network contained more than 20 modules, the module pairs with the most highly correlated ME were iteratively merged until 20 modules remained. Each expressed gene was then reassigned to the module to which it is most highly correlated to the ME (referred to as the gene's module membership, or kME). Genes with maximum kME <0.4 were left unassigned (defined as module 0), as we have done in previous analyses (*Hawrylycz et al., 2012*). Since networks are unaffected by changes in labelling, we then re-labeled modules by percent of neuron-enriched genes so that those with the highest percentage of neuronal markers (*Zhang et al., 2014*) have lower numbers and those with the lowest percentage have higher numbers, as described previously (*Hawrylycz et al., 2012*).

We compared ME expression with 24 pathological and demographic measures, and used SVA to assess significance, defined here as Bonferroni-corrected p<0.05. In addition to the modules discussed in the Results, we found a single module of Y-chromosome genes in each network with nearly exclusive expression in males, as expected.

## Gene set comparison between studies

In order to compare our differential expression results with prior work, we first assembled lists of genes differentially expressed between donors diagnosed with AD and matched controls from nine previous studies (*Colangelo et al., 2002*; *Blalock et al., 2004*; *Liang et al., 2008*; *Webster et al., 2009*; *Avramopoulos et al., 2011*; *Miller et al., 2013*; *Zhang et al., 2013*; *Satoh et al., 2014*; ). *Figure 6—source data 4* describes in more detail specifically how we derived

each gene list. *Figure 6—source data 5* includes all gene lists. We then sorted and ranked all genes in our analysis with respect to fold difference (for dementia vs. control) or correlation with RIN in hippocampus and temporal cortex, scaling from 0 to 1. We then noted the ranks of external gene lists in our sorted lists, including the 25th, 50th, and 75th percentile values. We calculated p-values using a two-sided Wilcoxon rank sum test to measure divergence from a random distribution, with the R function 'wilcox.test' (*Bauer, 1972*). This schema is a modification of one previously described (*Miller et al., 2013*).

We compared the relationships between gene expression and local quantifications of tau pathology in our study with one previous study (*Blalock et al., 2004*). Correlations between gene expression (defined by the probe for each gene with maximal expression) and 'NFT Score' were calculated for the comparison study using publicly available data (GEO: GSE1297). Correlations between gene expression and AT8 IHC quantification from fresh frozen tissue were calculated in this study and the resulting correlations were themselves correlated for comparison.

Enrichment for gene ontology (GO) categories was performed using ToppFun with default parameters, which is available as part of the ToppGene Suite (*Chen et al., 2009*) (https://toppgene. cchmc.org/). Cell type-specific expression levels were collected from a published data set of selective expression in human neurons, astrocytes, microglia, and oligodendrocytes (*Zhang et al., 2014*) (http://web.stanford.edu/group/barres_lab/brain_rnaseq.html). Cell type enrichment was calculated by comparing gene lists in this study against genes with 2-fold enrichment in one verses all other cell types and FPKM >1. Cell type enrichment was largely confirmed by comparison with mouse-derived gene sets downloaded from NeuroExpresso (neuroexpresso.org) (*Mancarci et al., 2016*) using the same strategy.

## Data and software availability

All images and most data presented in this manuscript are freely available from the resource website, http://aging.brain-map.org/. Code and remaining files required to reproduce all analyses and associated figure panels are available as part of the Github repository (https://github.com/AllenInstitute/agedbrain; *Miller, 2017*; copy archived at https://github.com/elifesciences-publications/agedbrain). Raw RNA-Seq data (FASTQ) and the output files after alignment (bam/FASTQ) are available for controlled access at NIAGADS: https://www.niagads.org/datasets/ng00059. TbT-normalized data (both before and after controlling for RIN) are also available through GEO (GSE104687).

## Additional resources

Technical documentation describing the ACT cohort, all experimental procedures (i.e., tissue collection, tissue processing, quantitative data generation), and weighted analysis in more detail are freely available at http://help.brain-map.org/display/aging/Documentation.

## Acknowledgements

We thank the Allen Institute for Brain Science founders, PG Allen and J Allen, for their vision, encouragement, and support. This work was funded by a grant to CD Keene, RG. Ellenbogen and Ed Lein from the Paul G Allen Family Foundation, and supported by National Institutes of Health grants U01AG006781 andP50AG005136, and the Nancy and Buster Alvord Endowment. We are grateful for the technical and administrative support of the staff members in the Allen Institute and from the University of Washington and Kaiser Permanente Washington Health Research Institute who are not part of the authorship of this paper. ROS/MAP study data were provided by the Rush Alzheimer's Disease Center, Rush University Medical Center, Chicago. Data collection was supported through funding by NIA grants P30AG10161, R01AG15819, R01AG17917, R01AG30146, R01AG36836, U01AG32984, U01AG46152, the Illinois Department of Public Health, and the Translational Genomics Research Institute. Mayo RNA-Seq study data were provided by the following sources: The Mayo Clinic Alzheimers Disease Genetic Studies, led by Dr. Nilufer Taner and Dr. Steven G. Younkin, Mayo Clinic, Jacksonville, FL using samples from the Mayo Clinic Study of Aging, the Mayo Clinic Alzheimers Disease Research Center, and the Mayo Clinic Brain Bank. Data collection was supported through funding by NIA grants P50 AG016574, R01 AG032990, U01 AG046139, R01 AG018023, U01 AG006576, U01 AG006786, R01 AG025711, R01 AG017216, R01 AG003949, NINDS grant R01 NS080820, CurePSP Foundation, and support from Mayo Foundation. Study data includes samples

collected through the Sun Health Research Institute Brain and Body Donation Program of Sun City, Arizona. The Brain and Body Donation Program is supported by the National Institute of Neurological Disorders and Stroke (U24 NS072026 National Brain and Tissue Resource for Parkinsons Disease and Related Disorders), the National Institute on Aging (P30 AG19610 Arizona Alzheimers Disease Core Center), the Arizona Department of Health Services (contract 211002, Arizona Alzheimers Research Center), the Arizona Biomedical Research Commission (contracts 4001, 0011, 05–901 and 1001 to the Arizona Parkinson's Disease Consortium) and the Michael J Fox Foundation for Parkinsons Research. MSBB data were generated from postmortem brain tissue collected through the Mount Sinai VA Medical Center Brain Bank and were provided by Dr. Eric Schadt from Mount Sinai School of Medicine.

## Additional information

### Funding

| Funder | Grant reference number | Author |
|---|---|---|
| National Institutes of Health | P50AG005136 | Eric B Larson<br>Paul K Crane |
| National Institutes of Health | U01AG006781 | Eric B Larson<br>Paul K Crane |
| Nancy and Buster Alvord Endowment | | Richard G Ellenbogen<br>C Dirk Keene<br>Ed Lein |

The funders had no role in study design, data collection and interpretation, or the decision to submit the work for publication.

### Author contributions

Jeremy A Miller, Formal analysis, Investigation, Methodology, Writing—original draft, Writing—review and editing; Angela Guillozet-Bongaarts, Formal analysis, Supervision, Methodology, Project administration; Laura E Gibbons, Steve Horvath, Shubhabrata Mukherjee, Formal analysis; Nadia Postupna, Formal analysis, Investigation, Project administration; Anne Renz, Shannon E Rose, Chris Barber, Darren Bertagnolli, Kristopher Bickley, Krissy Brouner, Shiella Caldejon, Mindy L Chua, Natalie M Coleman, Eiron Cudaback, Christine Cuhaciyan, Nadezhda I Dotson, Garrett Gee, Terri L Gilbert, Fiona Griffin, Caroline Habel, Zeb Haradon, Nika Hejazinia, Leanne L Hellstern, Kim Howard, Robert Howard, Justin Johal, Nikolas L Jorstad, Samuel R Josephsen, Florence Lai, Eric Lee, Tracy Lemon, Desiree A Marshall, Jose Melchor, Julie Pendergraft, Lydia Potekhina, Elizabeth Y Rha, Samantha Rice, David Rosen, Abharika Sapru, Emily Sherfield, Shu Shi, Michael Tieu, Investigation; Allison E Beller, Formal analysis, Project administration; Susan M Sunkin, Aimee Schantz, Elaine Shen, Project administration; Lydia Ng, Formal analysis, Supervision, Methodology; Kimberly A Smith, Supervision, Methodology, Project administration; Aaron Szafer, Rachel A Dalley, Formal analysis, Methodology; Mike Chapin, Nathalie Gaudreault, Andy J Sodt, Supervision, Investigation; Nick Dee, Jeff Goldy, Chihchau L Kuan, Formal analysis, Investigation, Methodology; Tsega Desta, Formal analysis, Supervision; Tim A Dolbeare, Supervision, Investigation, Methodology; Michael Fisher, Xianwu Li, Nivretta Thatra, Formal analysis, Investigation; Felix Lee, Julie Nyhus, Angela M Wilson, Investigation, Methodology; Thomas J Montine, Conceptualization, Methodology; Eric B Larson, Conceptualization, Supervision; Amy Bernard, Formal analysis, Supervision, Investigation, Methodology; Paul K Crane, Conceptualization, Supervision, Writing—original draft, Writing—review and editing; Richard G Ellenbogen, Conceptualization, Supervision, Writing—review and editing; C Dirk Keene, Conceptualization, Supervision, Investigation, Methodology, Writing—original draft, Project administration, Writing—review and editing; Ed Lein, Conceptualization, Formal analysis, Supervision, Methodology, Writing—original draft, Writing—review and editing

### Author ORCIDs

Jeremy A Miller ⓘ http://orcid.org/0000-0003-4549-588X
Laura E Gibbons ⓘ http://orcid.org/0000-0001-5054-2543

Nadezhda I Dotson (iD) http://orcid.org/0000-0003-3414-3176
Eric Lee (iD) http://orcid.org/0000-0002-7166-0909
Julie Pendergraft (iD) http://orcid.org/0000-0002-6066-4576
Amy Bernard (iD) http://orcid.org/0000-0003-2540-1153
C Dirk Keene (iD) https://orcid.org/0000-0002-5291-1469
Ed Lein (iD) http://orcid.org/0000-0001-9012-6552

**Decision letter and Author response**
Decision letter https://doi.org/10.7554/eLife.31126.035
Author response https://doi.org/10.7554/eLife.31126.036

## Additional files
### Supplementary files
• Transparent reporting form
DOI: https://doi.org/10.7554/eLife.31126.030

### Major datasets
The following datasets were generated:

| Author(s) | Year | Dataset title | Dataset URL | Database, license, and accessibility information |
|---|---|---|---|---|
| Miller JA, Keene CD, Lein E | 2017 | Aging, Dementia, and TBI Study | https://www.niagads.org/datasets/ng00059 | Publicly available at the National Institute on Aging Genetics of Alzheimer's Disease Data Storage Site (accession no. ng00059) |
| Miller JA, Keene CD, Lein ES | 2017 | Aging, Dementia, and TBI Study | https://www.ncbi.nlm.nih.gov/geo/query/acc.cgi?acc=GSE104687 | Publicly available at the NCBI Gene Expression Omnibus (accession no: GSE104687) |

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
