## [Decision Letter]

[Editors’ note: a previous version of this study was rejected after peer review, but the authors submitted for reconsideration. The first decision letter after peer review is shown below.]

Thank you for submitting your work entitled "Neuropathological and transcriptomic characteristics of the aged brain" for consideration by *eLife*. Your article has been reviewed by three peer reviewers, and the evaluation has been overseen by a Reviewing Editor and a Senior Editor. We apologize for the delay in returning this decision to, but there was an extended period of discussion among the reviewers.

Our decision has been reached after consultation between the reviewers. Based on these discussions and the individual reviews below, we regret to inform you that the present manuscript will not be considered further for publication in *eLife*.

Although we recognize that the study represents a large and potentially valuable resource for the community, we are declining the present version of the manuscript because the reviewers found it difficult to evaluate the precise reasons for the discrepancies between the present negative results and those obtained from other carefully carried out prior studies. It is *eLife* policy that all data and analysis code needed to derive the analysis results from the data should be made available. It is possible that if the details (including code) of the analyses are made clearer in a revised version that the reasons for the discrepancies will be clearer and that we will be able to make a more informed decision about publication of the manuscript. Hence, despite the rejection, we encourage a resubmission, especially if the points raised by the reviewers can be addressed. Note that complete agreement between prior studies and this one is not required, but in the face of major disagreements, the present study must help to illuminate the cause of the discrepancies and should add to the ability of careful readers to come to a more integrated understanding. In the hopes of making the issues raised as clear as possible we are including both the original reviews and additional relevant comments made during discussion of the reviews.

Reviewer #1:

Considering this work as a data resource, my main consideration is in how well the results can be reproduced and repurposed, for the many other researchers who will be interested in replicating (or not) findings in their own cohorts. Since the conclusions of this study are somewhat at odds with several other studies, it is particularly important to make all the data needed for reprocessing available.

In particular, since many people might want to use their own normalization and RIN was emphasized in this paper, that would be an important variable to share, but I was unable to find that variable in the data-sharing website (nor could I find PMI). In addition, it would be helpful to provide other information that people would ostensibly use in a normalization – percentage aligned reads or other variables that you had access to that you ultimately chose not to use in this particular normalization such as number of ribosomal bases. Also, batch data was unavailable. If these are available and I'm just missing them, then this criticism dissolves.

However, it does raise a related point. The exact scripts used in your normalization will enable others to understand how additional variable change their results, in addition to the substantial time savings from having to figure out where everything is in the different files and trying to exactly infer your code from the text. Specifically, it should be possible to take the input files shared on the web – which is all any external researcher will be able to access – and arrive at your normalized data. Indeed, these are essential to providing a firm foundation for alternative normalizations of the data, which I suspect is going to be a major interest of many researchers given the particular results of this study.

I appreciate that you have provided your normalized output, but to actually consider modifications to the normalization, the scripts used in getting there are essential. There is a note that code for figures is available on GitHub – however I don't have a link for that and it's not clear it covers normalization. If so – great! – but I don't have a way of looking into this.

Reviewer #2:

The authors present an analysis of the ACT cohort of human brain tissue taken from aged individuals with and without various pathologies (AD, dementia and TBI). They analyze a broad range of assays such as protein markers and RNA-seq from several brain regions. While these efforts are impressive, in general it seems the findings primarily confirm what is already known about relationships between markers such as AB, pTau, inflammation, and aging, AD or dementia. While a new category (M16) of co-expressed genes was found to be correlated with pTau load, it is not clear whether this is particularly novel in relation to aging or dementia, or is informative for understanding the etiology of either. The paper initially seemed geared towards understanding the aging brain, but very little is presented to shed light on aging and most of what is presented regarding dementia and AD does not appear to be new. It is possible the study contains novel findings that were lost on this reader, but clearer descriptions and context are needed to make this apparent.

The investigation into RNA quality correlated with dementia is interesting and important, but seems unfinished. The finding that previous studies may be flawed due to a lack of accounting for RNA quality maybe significant enough for a short report. However, it also seems to undermine the current dataset. Given the discussion of brain pH and the greatest discordance between this study and a study which took this factor into account raises questions about the relationship between RIN and factors like pH. It is not clear from the analysis presented here that RIN is a sufficiently representative metric to account for all necessary variables. From the data in this study there appeared to be at least two potential issues in differential expression (DE) analysis. (1) Decreased RNA quality correlated with dementia but also (2) increased variance with age. The second factor was mentioned in the Discussion section but not clearly described in the Results section and so it is not clear how this may have impacted DE analysis. Thus, determining the correlation between RIN and dementia is an important issue, it is not clear that the authors are presenting an actual solution.

The main take-away seems to be that additional work is needed to optimize RNA isolation from pathological brain samples and perhaps a more in-depth investigation into increased expression variability with age. While decreased RNA quality and increase variance may be indicative of biological changes, this study does not seem to help clarify these possibilities. Improved methods to either remove or account for these factors might open up other possible lines of investigation. For example, the disconnect between pathological markers and dementia was interesting but only superficially investigated. Assuming the hurdles of RNA quality and high variance can be overcome, it would be a natural follow up to probe for gene expression signatures of "tolerance". While I agree that the ACT cohort seems to be a valuable resource, the limitations highlighted in this analysis make it unclear exactly what new information we might pull from it and how best to pull the information.

Reviewer #3:

This work represents an observational study that looks to characterize and associate various factors associated with Alzheimer's dementia (pathological markers, inflammatory markers, gene expression data and clinical dementia scores) by completing various measurements in brain specimens for which RNA sequencing data is publicly available through the ACT study (107 donors). Compared to prior studies, they evaluate an overall older sample population. They find an expected correlation between p-Tau levels and dementia. They find a positive correlation between astrocyte and microglia-related gene expression and hippocampal p-Tau levels, but no significant correlation with inflammatory related metrics and dementia.

The authors do not find a significant correlation between any gene expression network and dementia or other metric, which is troubling and pose serious doubts about the data – sample, processing, QC (batch effects), RNA-seq and downstream analyses. The most striking result that the authors present is the absence of any differentially expressed genes between AD and control samples in both hippocampus and cortex, which the authors agree is "surprising. One reason for this is likely the use of SVA which removes surrogate variables that may be confounders, or may not, especially if they are related to sources of variation like cell type. For example, we know at some stages that neurons are down, and there is astrogliosis, as the latter is a pathological feature. This signal has likely been removed by SVA. SVA, although widely used for some reason, has many limitations (e.g. Nature Methods 14, 218-219(2017)doi:10.1038/nmeth.4190). The authors should explore the major PCs in the data, understand what technical or biological factors they are related to. Removing technical confounders is important, but biological "confounders" may be of interest.

Unsurprisingly (as they did not find any DE genes) after performing WGCNA the authors found no modules (module-eigenegenes) to be correlated with diagnosis or AD pathological traits. Many publications have found hundreds of significantly differentially expressed genes between AD and control postmortem human samples (Colangelo et al., 2002, Blalock et al., 2004, Liang et al., 2008, Webster et al., 2009, Avramopoulos et al., 2011, Miller et al., 2013, Zhang et al., 2013, Narayanan et al. 2014, Satoh et al., 2014). The authors claim that many studies do not account for RIN quality, unlike this study, and as a result find gene expression changes between AD cases and controls. However, there are several issues with this:

a) It is unclear why there is discrepancy between the results from published studies which do account for RIN quality (like Zhang el al., 2013, Narayanan et al., 2014, etc. as they use linear regression to remove the effects of RIN) and this data? In this respect can the authors plot the logFC between AD and controls from their data and compare that with published data like Zhang et al. and Narayanan et al.?

b) It seems that the RIN values are heavily confounded with diagnosis in this study and removing the effect of RIN values using regression actually removes the signal. In fact, the authors claim "we find a substantially lower RNA quality in dementia cases vs. controls in all four brain regions". This is a major batch effect and technical confounder that limits the entire value of this study. In the end, the almost total confounding of low RIN with dementia status seriously undermines the conclusions that RIN is driving the changes.

Can the authors show distribution of RIN values (histogram) separately for controls and AD samples? Also, the authors can use the first 5 PCs of gene-expression (before and after regression – separately) and correlate them with various traits like diagnosis, age, sex, PMI, sequencing biases and most importantly RIN.

c) Can they take a subset of the study that is adequately powered relative to previous studies, where cases and controls are matched for RIN and show they get the same result?

d) It is also possible that the systematically lower RIN in demented cases resulted in sequencing biases like AT/GC bias, duplication rate, etc. but the authors seem to have not taken that into account while doing the analysis. Previous publications have shown that these sequencing biases are much better predictor of RNA quality than RIN itself. (Feng et al., 2015)

e) This is not a trivial issue that the authors do not find any DE genes, just based on the huge changes in cell-type proportion between AD and controls (neuronal loss and gliosis), we expect many differentially expressed genes in AD. In fact, the authors show huge microgliosis (Iba-1 staining, Figure 4), but cannot explain any lack of change in differential expression of any microglia related genes. Thus, the lack of any DE genes raises serious doubts about the analysis.

f) In fact, the widespread pathology in the AD samples as shown by AT8 staining, Nissl staining does not correlate with lack of any DE genes. Do the authors expect that pathological dementia cases have no changes in gene expression, – in this case what possibly accounts for the pathology?

Reviewer 3:

The two main issues can be summarized as:

1) there is a tendency just to apply hidden covariate correction models such as SVA without much introspection and SVA is particularly problematic.

2) confounding of RIN with case control status.

This work while perhaps well done to some degree, is not properly interpreted. Use of SVA as it is applied here, almost certainly removes biological signal. The authors must spend some time better understanding what PCs/SVs have been removed, what they are related to -- this will lead to a very different set of conclusions, I suspect.

[Editors’ note: what now follows is the decision letter after the authors submitted for further consideration.]

Thank you for submitting your article "Neuropathological and transcriptomic characteristics of the aged brain" for consideration by *eLife*. Your article has been reviewed by three peer reviewers, and the evaluation has been overseen by Sacha Nelson as the Reviewing Editor and Eve Marder as the Senior Editor. The 3 reviewers have opted to remain anonymous.

The reviewers have discussed the reviews with one another and the Reviewing Editor has drafted this decision to help you prepare a revised submission.

Summary:

The authors present a large multi modal analysis of brain specimens from a prior RNA sequencing study of 107 aged human donors including many with dementia and traumatic brain injury. The major findings of the study are negative with respect to associations between neuroinflammation and aging or dementia, and between these factors and gene expression. The paper suggests that RNA quality was correlated with dementia status limiting the ability to independently assess correlations between dementia and gene expression. Although there was disagreement between the reviewers, on balance the reviewers and editors felt that the value of the resource to the community, and the value of the cautions raised with respect to RNA quality, outweighed the lack of other clear findings.

Essential revisions:

All three reviews are included for the author's benefit, but only the major points raised by reviewer #3 and the minor points raised by reviewer's #2 need to be addressed in the revision. Please indicate how these have been addressed in the letter accompanying the revision. These changes should only require textual changes, addition of a supplemental table or figure, and further notes about how code used in analyses were deposited.

Reviewer #1:

I appreciate the authors' effort in making the code available, and in added/reanalysis to demonstrate that the initial finding of little to no association between specific genes and AD pathology were not simply due to the computational method. While this rules out the simplest explanation for the discrepancy between this study and others, it does not in my opinion address the issue fully enough. As the author's state the discrepancy could be due to biological or technical differences in the data set. I don't think ruling out the analytical method as the source of the discrepancy is sufficient to "illuminate the cause of the discrepancies" nor does it sufficiently "add to the ability of careful readers to come to a more integrated understanding" of this study in the context of other carefully carried out prior studies.

I don't mean to imply that I wanted the authors to figure out a way to make their data fit with previous studies, but rather to say that the study still feels unfinished. The finding that their cohort agrees with other studies in some ways but not in others is interesting, and the finding that RNA quality is correlated with AD pathology is also important. But, I don't see any significant light shed on why. I also don't see a straightforward way to address this question in the short time period expected for a resubmission. I am certainly open to hearing the thoughts/suggestions of the other reviewers in case they see some potential that I'm missing. If not, for me, it boils down to a correlation that exists in some studies (including theirs), and unexplained major discrepancies, which doesn't seem suitable for *eLife*.

Reviewer #2:

The authors have adequately addressed my previously expressed concerns regarding both the specific findings of this paper and the ability to reproduce them, through their inclusion of code and tables used in their analysis, and as long as these are included with the publication or in GitHub. There is definite benefit in releasing this analysis, as it is reasonably performed and now well-documented.

Reviewer #3:

In the current revision, the authors have thoroughly addressed concerns from the reviewers. The major concern raised by all the reviewers is that SVA most likely removed biological signal from the dataset. The authors need to more clearly state this in the final manuscript, and acknowledge the likely many issues with their data – this work does not convincingly undermine previous well controlled studies and that should also be stated. But, the data will be useful to the field as it is a large data set, and the authors have done a good job responding to the reviewers' concerns, so I am favorably inclined.

a) The authors have now looked at the pre-regressed data and found that PC1 is correlated with diagnosis in all brain regions except FWM (where PC2 is correlated to diagnosis). However, the PC1 in these cases is also correlated with RIN, which is a huge confounder that limits the entire analysis. So, unsurprisingly accounting for RIN removes all the signal. The authors should make a point to report the PC correlations with various biological variable like diagnosis, age, sex, RIN, etc. as a supplemental table or supplemental figure pre-regression and post-regression.

b) To address the major confounding issue, the authors take a subset of the study (that is adequately powered relative to previous studies) where cases and controls are matched for RIN, they get the same result – no significant DE genes between cases and controls. Moreover, the authors acknowledge that the lack of any gene-expression changes given severe inflammation and Tau pathology (AT8 staining) in the samples is perplexing. This is very surprising especially given the consistency of previous studies at the RNA and more recently in the AMP-AD data, at the protein level.

My general sense is that the current results are due to confounders, as well as the fact that the underlying biology involves changes in cell composition, which when corrected for, removes the major signal. However, in light of the thorough analysis of the dataset using multiple statistical methods, and its size, this reviewer feels that the dataset/manuscript should be made publicly available so other can use and address the issues with the dataset. As a condition of publication, the authors should set up a GitHub or similar code-deposit service and deposit all the analysis done for this study in a systematic manner. Moreover, the metadata and the pre-normalized and SVA regressed gene-expression data should be submitted to GEO in addition to raw fastqs which should be deposited to dbGAP or similar services

---

## [Author Response]

[Editors’ note: the author responses to the first round of peer review follow.]

Reviewer #1:Considering this work as a data resource, my main consideration is in how well the results can be reproduced and repurposed, for the many other researchers who will be interested in replicating (or not) findings in their own cohorts. Since the conclusions of this study are somewhat at odds with several other studies, it is particularly important to make all the data needed for reprocessing available.

We completely agree about the importance of reproducibility and repurposing, and that the reviewer for bringing up this issue. For that reason all data, code, and additional files required to reproduce all analyses available in the manuscript are now freely and publicly available, either at http://aging.brain-map.org or as part of a zip file submitted with this version of the manuscript. In addition, the raw data files for RNA-Seq is now available for controlled access through NIAGADS (https://www.niagads.org/datasets/ng00059).

In particular, since many people might want to use their own normalization and RIN was emphasized in this paper, that would be an important variable to share, but I was unable to find that variable in the data-sharing website (nor could I find PMI). In addition, it would be helpful to provide other information that people would ostensibly use in a normalization – percentage aligned reads or other variables that you had access to that you ultimately chose not to use in this particular normalization such as number of ribosomal bases. Also, batch data was unavailable. If these are available and I'm just missing them, then this criticism dissolves.

RIN and batch are available as part one the columns information file that available when downloading the RNA-Seq data from the “Download” tab at http://aging.brain-map.org, and we apologize for not making the location of this information clearer in the initial version of our manuscript. PMI is not available; however, as all of these cases are rapid autopsy cases, the PMI for every donor was <8 hours.

However, it does raise a related point. The exact scripts used in your normalization will enable others to understand how additional variable change their results, in addition to the substantial time savings from having to figure out where everything is in the different files and trying to exactly infer your code from the text. Specifically, it should be possible to take the input files shared on the web – which is all any external researcher will be able to access – and arrive at your normalized data. Indeed, these are essential to providing a firm foundation for alternative normalizations of the data, which I suspect is going to be a major interest of many researchers given the particular results of this study.

We had originally intended to make all code available upon publication as part of a GitHub repository and mentioned this in the original text, but we recognize that we should have made it available during the review process. This code is now available to reviewers as a zip file. We agree that reproducibility is particularly important with regards to this normalization strategy and would encourage others replicate our analysis, and to analyze or normalize the data in a different way.

I appreciate that you have provided your normalized output, but to actually consider modifications to the normalization, the scripts used in getting there are essential. There is a note that code for figures is available on GitHub – however I don't have a link for that and it's not clear it covers normalization. If so – great! – but I don't have a way of looking into this.

The code covers all of the analyses, including normalization, and is now available to the reviewers.

Reviewer #2:The authors present an analysis of the ACT cohort of human brain tissue taken from aged individuals with and without various pathologies (AD, dementia and TBI). They analyze a broad range of assays such as protein markers and RNA-seq from several brain regions. While these efforts are impressive, in general it seems the findings primarily confirm what is already known about relationships between markers such as AB, pTau, inflammation, and aging, AD or dementia. While a new category (M16) of co-expressed genes was found to be correlated with pTau load, it is not clear whether this is particularly novel in relation to aging or dementia, or is informative for understanding the etiology of either. The paper initially seemed geared towards understanding the aging brain, but very little is presented to shed light on aging and most of what is presented regarding dementia and AD does not appear to be new. It is possible the study contains novel findings that were lost on this reader, but clearer descriptions and context are needed to make this apparent.

We appreciate the reviewer’s understanding of the work that went into creating this resource, and of the insightful critiques. The analyses presented in this manuscript with respect to aging and dementia are largely confirmatory in nature, as the reviewer indicates. Our intension was to present a resource that includes representative features and pathologies of the aged brain, rather than to specifically target changes in the brain that occur during aging (since we do not have young donors in this cohort). By including confirmatory analyses, our goal was to show the validity of the various components of our study. With regards to novelty, to the best of our knowledge we are the first to show a direct connection between genes associated with dementia and RNA quality.

The investigation into RNA quality correlated with dementia is interesting and important, but seems unfinished. The finding that previous studies may be flawed due to a lack of accounting for RNA quality maybe significant enough for a short report. However, it also seems to undermine the current dataset.

We thank the reviewer for these thoughts about our findings with respect to previous reports. Upon further evaluation of the literature, we have scaled back on discussions about how previous studies may be flawed, and instead focus on the importance of accounting for RNA quality. The overlap in genes related to RNA quality and dementia suggests that RNA quality genes could be mistaken for dementia genes if not properly controlled for.

Given the discussion of brain pH and the greatest discordance between this study and a study which took this factor into account raises questions about the relationship between RIN and factors like pH. It is not clear from the analysis presented here that RIN is a sufficiently representative metric to account for all necessary variables.

The question of how best to measure RNA quality is an interesting one, but one that we feel is relatively minor when compared against the relationship between RNA quality and dementia. We now include Table 2 which shows that donors with AD have lower RNA quality than non-demented controls in several studies, in some cases based on RIN and in other cases based on pH, suggesting that this relationship holds for both measures of RNA quality. For this study, pH is not available, but considering RIN can explain up to 80% of gene expression variation for some genes, RIN seems to be a good proxy for RNA quality.

From the data in this study there appeared to be at least two potential issues in differential expression (DE) analysis. (1) Decreased RNA quality correlated with dementia but also (2) increased variance with age. The second factor was mentioned in the Discussion section but not clearly described in the Results section and so it is not clear how this may have impacted DE analysis. Thus, determining the correlation between RIN and dementia is an important issue, it is not clear that the authors are presenting an actual solution.

We did not intend to state that gene expression has higher variance with increasing age, and apologize for the confusion. Instead, we find that the relationship between dementia diagnosis and pathology (or gene expression) is less defined in the oldest old. This means that pathology (or genes) that are expected to show differential patterns in younger donors with and without dementia may not show these differential patterns in an aged cohort. Both this effect as well as the overlapping set of genes associated with dementia and RNA-quality could have led to our findings with respect to differential expression between donors with and without dementia.

The main take-away seems to be that additional work is needed to optimize RNA isolation from pathological brain samples and perhaps a more in-depth investigation into increased expression variability with age. While decreased RNA quality and increase variance may be indicative of biological changes, this study does not seem to help clarify these possibilities. Improved methods to either remove or account for these factors might open up other possible lines of investigation.

We appreciate the reviewer’s feedback with respect to potential changes in experimental design, but respectfully disagree about specifics in methodology negatively impacting our results. The difference in RNA quality between normal and pathological samples is more likely to be a biological effect or something to do with differences in agonal state, than an issue of optimizing RNA isolation, and now include a table showing these differences in several additional cohorts. Having said this, methods for RNA isolation that are more robust to RNA degradation could dramatically improve future studies of the aged brain. In addition, we do not see increased gene expression variability with age (see previous response), and have attempted to update the text accordingly.

For example, the disconnect between pathological markers and dementia was interesting but only superficially investigated. Assuming the hurdles of RNA quality and high variance can be overcome, it would be a natural follow up to probe for gene expression signatures of "tolerance". While I agree that the ACT cohort seems to be a valuable resource, the limitations highlighted in this analysis make it unclear exactly what new information we might pull from it and how best to pull the information.

This is a great insight, and we agree that identifying gene signatures of tolerance would be an important next step. In fact, we are currently pursuing this as an additional study beyond the scope of this manuscript. We hope that by making this resource and our analysis freely available (see responses to Reviewer #1), additional insights into dementia and the aged brain could be obtained.

Reviewer #3:This work represents an observational study that looks to characterize and associate various factors associated with Alzheimer's dementia (pathological markers, inflammatory markers, gene expression data and clinical dementia scores) by completing various measurements in brain specimens for which RNA sequencing data is publicly available through the ACT study (107 donors). Compared to prior studies, they evaluate an overall older sample population. They find an expected correlation between p-Tau levels and dementia. They find a positive correlation between astrocyte and microglia-related gene expression and hippocampal p-Tau levels, but no significant correlation with inflammatory related metrics and dementia.The authors do not find a significant correlation between any gene expression network and dementia or other metric, which is troubling and pose serious doubts about the data – sample, processing, QC (batch effects), RNA-seq and downstream analyses. The most striking result that the authors present is the absence of any differentially expressed genes between AD and control samples in both hippocampus and cortex, which the authors agree is "surprising. One reason for this is likely the use of SVA which removes surrogate variables that may be confounders, or may not, especially if they are related to sources of variation like cell type. For example, we know at some stages that neurons are down, and there is astrogliosis, as the latter is a pathological feature. This signal has likely been removed by SVA. SVA, although widely used for some reason, has many limitations (e.g. Nature Methods 14, 218-219(2017)doi:10.1038/nmeth.4190). The authors should explore the major PCs in the data, understand what technical or biological factors they are related to. Removing technical confounders is important, but biological "confounders" may be of interest.

We appreciate the reviewer’s accurate summary of our study and suggestions for potential reasons for our relatively limited associations between gene expression and pathology-related metrics. As noted in the initial version of the text, we agree that our negative result (failing to find gene associated with dementia status) is surprising and could potentially reflect biological or technical differences in this data set compared with previous data (e.g., older cohort, large range in RIN). To assess whether SVA may be removing important gene expression signal, we have repeated our analysis using several additional strategies including some that do not use surrogate variables. In all cases we find the same result, indicating that this is not an artifact of our choice of statistical method. Specifically we add to the Materials and methods section:

“To assess the robustness of our result that few if any genes are significantly associated with dementia (or AD) status after controlling for RIN, we performed additional analyses to quantify significance of gene expression with respect to dementia status, in all cases defining significance as p<0.05 after Bonferroni correction. First, we repeated our analyses using additional statistical tests including 1) two tailed student t-test, 2) ANOVA, 3) and limma (Ritchie et al., 2015), in all cases defining two groups based on dementia (or AD) status. Second, we repeated the SVA analysis described above on the RIN-normalized RNA-Seq data available for download from the website (which uses a slightly different normalization schema, as described above). Third, we performed principal component analysis independently on each region using all genes, and used SVA to assess whether any of the top 25 PCs showed significant associate with dementia. Finally, we subsampled our data set to 70 donors who are matched for RIN, sex, and dementia status and repeated the SVA analysis using data that is not RIN-corrected to determine whether our particular RIN-normalization strategy could be biasing our ability to identify genes associated with dementia. In all cases we found two or fewer total genes associated with dementia or AD, indicating that our negative result is not due to improper statistical assessment.”

While not indicated in the text, we should note that SVA and ANOVA produce identical results using our current set of parameters. Code for reproducing all of these analyses are now included in this submission as a zip file.

Unsurprisingly (as they did not find any DE genes) after performing WGCNA the authors found no modules (module-eigenegenes) to be correlated with diagnosis or AD pathological traits. Many publications have found hundreds of significantly differentially expressed genes between AD and control postmortem human samples (Colangelo et al., 2002, Blalock et al., 2004, Liang et al., 2008, Webster et al., 2009, Avramopoulos et al., 2011, Miller et al., 2013, Zhang et al., 2013, Narayanan et al. 2014, Satoh et al., 2014). The authors claim that many studies do not account for RIN quality, unlike this study, and as a result find gene expression changes between AD cases and controls. However, there are several issues with this:a) It is unclear why there is discrepancy between the results from published studies which do account for RIN quality (like Zhang el al., 2013, Narayanan et al., 2014, etc. as they use linear regression to remove the effects of RIN) and this data? In this respect can the authors plot the logFC between AD and controls from their data and compare that with published data like Zhang et al. and Narayanan et al.?

We thank the reviewer for these thoughts about our findings with respect to previous reports. After more carefully reviewing the literature we have scaled back on our claims that differences with previous studies are likely due to earlier studies not accounting for RNA quality because, as multiple reviewers correctly point out, some previous studies do control for RIN and still find significant genes. Having said this, one of the points of Figure 6 is that, despite the lack of significant genes in our data set, we do find good agreement with genes differentially expressed in previous studies and genes with the highest (albeit non-significant) fold change differences in our current study. This suggests that there is concordance between studies, although we agree that the lower fold changes and non-significant results in our current study are surprising. We have added an additional large study of AD that controls for RIN (Allen et al) and find that it is one of the studies that best agrees with our current results. Finally, we note that nearly all of these studies also show a significant overlap between dementia related genes identified in previous studies and genes associated with RNA quality in this study, even in cases where RIN is corrected. Our interpretation of this is that many of the same genes are associated with both processes, which is how we now present our result.

b) It seems that the RIN values are heavily confounded with diagnosis in this study and removing the effect of RIN values using regression actually removes the signal. In fact, the authors claim "we find a substantially lower RNA quality in dementia cases vs. controls in all four brain regions". This is a major batch effect and technical confounder that limits the entire value of this study. In the end, the almost total confounding of low RIN with dementia status seriously undermines the conclusions that RIN is driving the changes.

We respectfully disagree that the result of lower RNA quality in dementia cases compared with controls is a confounding factor and instead argue that this is a feature that is commonly found in studies of AD. To back this up, we now present Table 2, which shows that in 5 of 8 other AD data sets that we could find, the same significant relationship is seen. Specifically we write:

“This result suggested a direct link between RNA quality and dementia status. Indeed, we find a substantially lower RNA quality in dementia cases vs. controls in all four brain regions (Figure 6; Figure 6—figure supplement 1), and this difference was not related to the time between death and autopsy (PMI; less than 8 hours for all donors). We next repeated this comparison on data from four additional population-based cohorts as part of the AMP-AD knowledge portal (https://www.synapse.org/ampad; (Bennett et al., 2012a, Bennett et al., 2012b, Allen et al., 2016)), and compared these with previous reports (Colangelo et al., 2002, Preece and Cairns, 2003, Durrenberger et al., 2010, Zhang et al., 2014). In five of the eight additional data sets assayed, donors with AD had significantly lower RIN than donors without dementia (Table 2). Donors with AD had significantly higher RNA quality in only one study (Allen et al., 2016). Thus the link between dementia status and RNA-quality is a broader phenomenon that is not unique to the ACT cohort, but is also not ubiquitous.”

Can the authors show distribution of RIN values (histogram) separately for controls and AD samples?

We appreciate the suggestion to display this data in multiple formats for the purposes of understanding what is going on with respect to RNA quality. Therefore, here is the distribution of RIN values for controls and dementia donors in histogram form:

This shows the same result that is currently shown in Figure 6 (higher RIN in controls), and therefore we feel that it would be redundant to include as part of the manuscript.

Also, the authors can use the first 5 PCs of gene-expression (before and after regression – separately) and correlate them with various traits like diagnosis, age, sex, PMI, sequencing biases and most importantly RIN.

As requested we have performed a comparison between the top 25 PCs (to explain a larger total fraction of the variance) and disease diagnosis on the RIN-controlled data, as described above, and found no PCs significantly related to dementia status. When we repeated this analysis using the data uncorrected for RIN we found exactly 1 PC corresponding to dementia in each brain region. This was the first PC in all brain regions except FWM, where it was the second PC (PC 1 in FWM likely corresponds to gene expression markers of white matter). These same PCs related with much more significant p-values to RIN (p~0 in all cases). We have expanded this PC analysis to other diagnostic traits and find that PCs significantly associated with tau in hippocampus, with sex, and with inflammation, both in the analyses before and after controlling for RNA quality, recapitulating the results described in out manuscript. We do not find any PCs associated with age at death.

Since this analysis is conceptually similar to what is done with WGCNA, in that vectors relating to large sources of variation are used in place of genes to reduce dimensionality for a comparison against sample metrics, we do not feel that the PC analysis adds significantly to the result of the paper, and currently do not include it. However, if the reviewer feels strongly about this subject, we could include it as a supplemental analysis. In either case, the code required to run this analysis is included as part of the submission.

c) Can they take a subset of the study that is adequately powered relative to previous studies, where cases and controls are matched for RIN and show they get the same result?

This is an excellent suggestion. We have done this and get the same result:

“Finally, we subsampled our data set to 70 donors who are matched for RIN, sex, and dementia status and repeated the SVA analysis using data that is not RIN-corrected to determine whether our particular RIN-normalization strategy could be biasing our ability to identify genes associated with dementia. In all cases we found two or fewer total genes associated with dementia or AD, indicating that our negative result is not due to improper statistical assessment.”

d) It is also possible that the systematically lower RIN in demented cases resulted in sequencing biases like AT/GC bias, duplication rate, etc. but the authors seem to have not taken that into account while doing the analysis. Previous publications have shown that these sequencing biases are much better predictor of RNA quality than RIN itself. (Feng et al., 2015)

Given that some genes show close to 80% of their variance explained by RIN, we find it unlikely that other features of RNA quality would better represent such biases. Having said this, we do find a bias in the relationship between a gene’s GC content and its correlation with RIN: genes with lower fractions of “C” nucleotide in their coding transcripts are more prone to decreased expression with RNA degradation.

**Author response image 2. respfig2:** 

This result was true independent of dementia status. We were unable to identify a strategy to account for these types of biases that led to results different from the ones presented in the manuscript.

e) This is not a trivial issue that the authors do not find any DE genes, just based on the huge changes in cell-type proportion between AD and controls (neuronal loss and gliosis), we expect many differentially expressed genes in AD. In fact, the authors show huge microgliosis (Iba-1 staining, Figure 4), but cannot explain any lack of change in differential expression of any microglia related genes. Thus, the lack of any DE genes raises serious doubts about the analysis.In fact, the widespread pathology in the AD samples as shown by AT8 staining, Nissl staining does not correlate with lack of any DE genes. Do the authors expect that pathological dementia cases have no changes in gene expression, – in this case what possibly accounts for the pathology?

These final points raised by the reviewer are arguably the most important. In some cases, we do find genes associated with specific pathology and that pathology is associated with dementia, as expected (e.g., tau). In other cases, we find genes associated with a pathology, but find that the pathology itself does not appear to be related to dementia status in our data set (e.g., inflammation/microglia, neuronal loss, gliosis). These findings (as well as the additional analyses not relying on SVA) indicate that the analysis itself is sound, but that our cohort for some reason does not show the usual relationship between disease diagnosis and pathology. Our best explanation for this result is due to the aged nature of this cohort, where in many cases disease diagnosis and pathology do not correlate. For example, eight of the 32 donors (25%) with severe NFT pathology (Braak stage >=5) and six of the 25 donors (24%) with severe amyloid pathology (CERAD score = 3) did not have dementia. It is possible that in such a cohort compensations are also in place for things like inflammation, gliosis, and neuronal loss. A final set of variables (e.g., amyloid load) do not find any associated genes, which is quite surprising. We don’t have a good explanation for this, but want to reiterate the soundness of our analysis.

Reviewer 3: The two main issues can be summarized as:1) there is a tendency just to apply hidden covariate correction models such as SVA without much introspection and SVA is particularly problematic.2) confounding of RIN with case control status.This work while perhaps well done to some degree, is not properly interpreted. Use of SVA as it is applied here, almost certainly removes biological signal. The authors must spend some time better understanding what PCs/SVs have been removed, what they are related to -- this will lead to a very different set of conclusions, I suspect.

We have discussed these two important points brought up by the reviewer above. In short, (1) repeating the analysis using different strategies leads to the same results, indicated that SVA is not negatively impacting our interpretation, and (2) we feel that the result that RIN is lower in dementia cases compared with controls is an important finding rather than a confounder. Having said this, we appreciate the reviewer bringing up these concerns so we can ensure that our analysis is as technically sound as possible.